# Maximal inhibitory effect of MOV10 on LINE-1 retrotransposition requires both the MOV10/LINE-1 association and granule formation

Qian Liu[1☯], Yaqi Liu[1☯], Yang Mao[2], Dongrong Yi[1], Quanjie Li[1], Jiwei Ding[1], Saisai Guo[1], Yongxin Zhang[1], Jing Wang[1], Jianyuan Zhao[1], Ling Ma[1], Xiaozhong Peng[3*], Shan Cen [1*], Xiaoyu Li[1*]

1 Institute of Medicinal Biotechnology, Chinese Academy of Medical Sciences and Peking Union Medical College, Beijing, China, 2 Department of Virus Research, Ningbo Municipal Center for Disease Control and prevention, Ningbo, China, 3 State Key Laboratory of Medical Molecular Biology, Department of Molecular Biology and Biochemistry, Institute of Basic Medical Sciences, Medical Primate Research Center, Neuroscience Center, Chinese Academy of Medical Sciences, School of Basic Medicine Peking Union Medical College, Beijing, China

☯ These authors contributed equally to this work as first author.
* pengxiaozhong@pumc.edu.cn (XP); shancen@imb.pumc.edu.cn (SC); lixiaoyu@imb.pumc.edu.cn (XL)

## Abstract

LINE-1 is the only active autonomous mobile element in the human, and its mobilization is tightly restricted by the host to maintain genetic stability. We recently reported that human MOV10 recruits DCP2 to decap LINE-1 RNA by liquid-liquid phase separation (LLPS), resulting in the inhibition of LINE-1 retrotransposition, while the detailed mechanism still awaits further exploration. In this report, we found that the extended motif II (563-675aa) and the C-terminal domain (907-1003aa) of MOV10 cooperated to achieve maximal inhibition on LINE-1 retrotransposition. The extended motif II involves the interaction between MOV10 and LINE-1, and the C-terminal domain is required for MOV10's association with G3BP1 and thereby the formation of granules. The association with LINE-1 through the extended motif II is dominantly attributed to MOV10-mediated anti-LINE-1 activity. On this basis, promoting large granules formation by the C-terminal domain warrants maximal inhibition of LINE-1 replication by MOV10. These data together shed light on the detailed mechanism underlying MOV10-mediated inhibition of LINE-1 retrotransposition, and provide further evidence supporting the important role of MOV10-driven granules in the anti-LINE-1 action.

## Author summary

Transposons, often referred to as mobile or jumping genes, are DNA sequences capable of replicating and integrating into various chromosomal locations, accounting for up to 56% of the human genome. Among these, LINE-1 (Long

---

**Data availability statement:** All data required for evaluating the conclusions of this study are available within the paper. Source data of the figures have been uploaded as Supporting information.

**Funding:** This work was supported by the National Natural Science Foundation of China (31870164 (XL), 32400139 (QL)), and the CAMS Innovation Fund for Medical Sciences (2021-I2M-1-038 (SC)). The funders had no role in study design, data collection and analysis, decision to publish, or preparation of the manuscript.

**Competing interests:** The authors have declared that no competing interests exist.

Interspersed Nuclear Element-1) stands out as the only autonomously active transposon in humans, with its activation being closely associated with the onset of numerous diseases. Consequently, the host tightly regulates its activity. In prior research, we demonstrated that the host factor MOV10 facilitates the degradation of LINE-1 RNA by recruiting the decapping enzyme DCP2 and forming phase-separated structures. Building on these findings, we have now pinpointed the critical domains within MOV10 essential for this function: 563-675aa and 907-1003aa. Our results reveal that the 563-675aa domain is responsible for mediating interactions with the LINE-1 ribonucleoprotein (RNP) complex, whereas the 907–1003aa domain plays a crucial role in granule formation. In conclusion, our results reveal that the anti-LINE-1 activity of MOV10 is contingent upon its interaction with the LINE-1 ribonucleoprotein (RNP) complex and its capacity to assemble cytoplasmic granules.

## Introduction

The full-length human long interspersed element 1 (LINE-1) is approximately 6,000 nucleotides in length and has three open reading frames: ORF1, ORF2 and ORF0 [1,2]. ORF1 protein (ORF1p) is a 40kDa nucleic acid-binding protein that functions as a chaperone [3,4]. ORF2 encodes a 149-kDa protein (ORF2p) with endonuclease and reverse transcriptase activities [5,6]. The function of ORF0 is not yet clear [2]. LINE-1 ORF1p, together with ORF2p, binds with LINE-1 RNA to form LINE-1 RNP in a cis-acting manner [7,8]. LINE-1 RNA reverses to cDNA and inserts into the genome through target-primed reverse transcription (TPRT), mediated by the ORF2 reverse transcriptase and endonuclease activities [5,9,10].

As the only active autonomous mobile DNA in the human genome, LINE-1 is a double-edged sword for the human genome [11]. Retrotransposition of LINE-1 is critical for species evolution, but for the individual it causes genetic instability by affecting the expression or regulation of nearby genes [11]. The insertion of LINE-1 can cause certain genetic diseases, such as pyruvate dehydrogenase complex deficiency [12–14]. Several lines of evidence showed that in tumor cells, both LINE-1 mRNA and proteins are present at high levels compared with normal cells, suggesting a close connection between LINE-1 and tumor occurrence and development [15–17].

Given that LINE-1 affects the host genome in so many ways, it is not surprising that cells have evolved various mechanisms to regulate its activity [18]. Numerous host factors have been found acting on different stages to restrict LINE-1 retrotransposition, such as SAMHD1 [19], TUT4/7 [20], RNASEH2 [21], SIRT6 [22] and RNA helicase MOV10 [23–25]. The cell factors control LINE-1 replication by multiple mechanisms [26], such as DNA methylation [26], chromatin condensation and nucleic acid editing [27].

Early studies showed that MOV10 inhibits the infectivity of multiple viruses, including human immunodeficiency virus, simian immunodeficiency virus, murine leukemia virus equine infectious anemia virus, hepatitis C virus and vesicular stomatitis virus

[28–31]. Later, we and others reported that MOV10 also inhibits retroelements including LINE-1 [24,32], through cooperation with several host factors such as DCP2, TUT4/7 and RNASEH2 [20,21,23,33].

MOV10 is a GTP-binding protein containing seven helicase motifs (I, Ia, II, III, IV, V and VI) [34], which may have ATPase/GTPase activity that can unwind DNA or RNA double strands. Helicases generally have seven conserved helicase domains and are classified into three superfamilies (SF-1, SF-2 and SF3), in addition to two subfamilies (F4 and F5) [35]. MOV10 was proposed as a putative superfamily-1 (SF-1) RNA helicase, with a conserved GxxxxGKT/S sequence in helicase motif I and a conserved DExx sequence in helicase motif II, which are also known as Walker A and Walker B [36]. It is known that motif I and II function as NTPase with binding to phosphate and $Mg^{2+}$ respectively, and other motifs bind to RNA or DNA [35].

Our previous study showed that MOV10 recruits DCP2 to form DCP2/MOV10/LINE-1 RNP complex (named DMLC) via liquid-liquid phase separation (LLPS), inducing efficient decapping of LINE-1 RNA, followed by LINE-1 RNA decay [33]. LLPS is a mechanism that governs the formation of membrane-less compartments in cells to meet biological requirements for spatiotemporal regulation [37,38]. In addition, stress granule marker proteins G3BP1 and TIA1 were shown to play a crucial role in assisting the formation of DMLC via LLPS [33]. Accordingly, the depletion of G3BP1 impaired MOV10-mediated decapping and degradation of LINE-1 RNA as well as the inhibition of LINE-1 retrotransposition [33]. These results together revealed a new strategy controlling LINE-1 mobilization by the host, while the detailed mechanism still awaits further exploration.

In this study, we identified the key motifs for MOV10 anti-LINE-1 activity using deletion mutants. We reported that extended motif II (amino acids 563-675) and the C-terminal domain (amino acids 907-1003) in MOV10 are critical for its inhibitory activity on LINE-1. Further evidence presented that the extended motif II (amino acids 563-675) in MOV10 is crucial for the interaction between MOV10 and LINE-1 RNP, and the C-terminal domain (amino acids 907-1003) is essential for its association with G3BP1 and thereby formation of granules.

## Results

### Helicase domain in MOV10 is essential for its anti-LINE-1 activity

To explore the detailed mechanism underlying MOV10-mediated restriction on LINE-1 replication, we first investigated the key motifs of MOV10 involved in the inhibition, using a set of MOV10 C-terminal truncations, in which the seven helicase motifs (I, Ia, II, III, IV, V, and VI) were removed sequentially (Fig 1A). Next, we assessed the effect of the MOV10 mutants on LINE-1 replication using a CMV-L1-neo^RT reporter as previously described [23]. In the CMV-L1-neo^RT reporter construct, a neomycin resistance gene containing an intron sequence was inserted between ORF2p and the 3'UTR of LINE-1 in the opposite direction [39]. Thus, the neomycin resistance gene can only be expressed upon successful reverse transcription of LINE-1 RNA, which can be determined by scoring G418-resistant cell colonies [40].

The retrotransposition assays were done in HeLa cells. Western blot of cell lysate showed a similar level of MOV10 expression in the cells transfected with the same amount of plasmids expressing either wild type or mutant forms of MOV10 (Fig 1B). The colony formation assay revealed a 90% reduction in the cell colonies with wild-type MOV10, compared with that of cells transfected with empty vector as a control (Fig 1C and 1D). Meanwhile, the number of cell colonies was reduced to a similar level ranged from approximate 40% to 50% in the presence of 1-906, 1-863, 1-727 or 1-675, about 70% for 1-644, while no significant effect was observed upon the overexpression of either 1-547, 1-523 or 1-92 (Fig 1C and 1D). This suggests that both amino acids 547-675 and 907-1003 of MOV10 are required for its inhibitory effect on LINE-1 replication, which consist of motif Ia, II and VI of RNA helicase, respectively. Accordingly, we found that once removal of motif VI, and further deletion until II, impaired the function of MOV10 to inhibit LINE-1 to a similar extent, about 20-30%, while a further deletion of motif Ia caused complete loss of anti-LINE activity (Fig 1E). The phased effect observed in Fig 1E suggests that the presence of both amino acids 547-675 and 907-1003 of MOV10 contribute to its maximal restriction to LINE-1 replication.

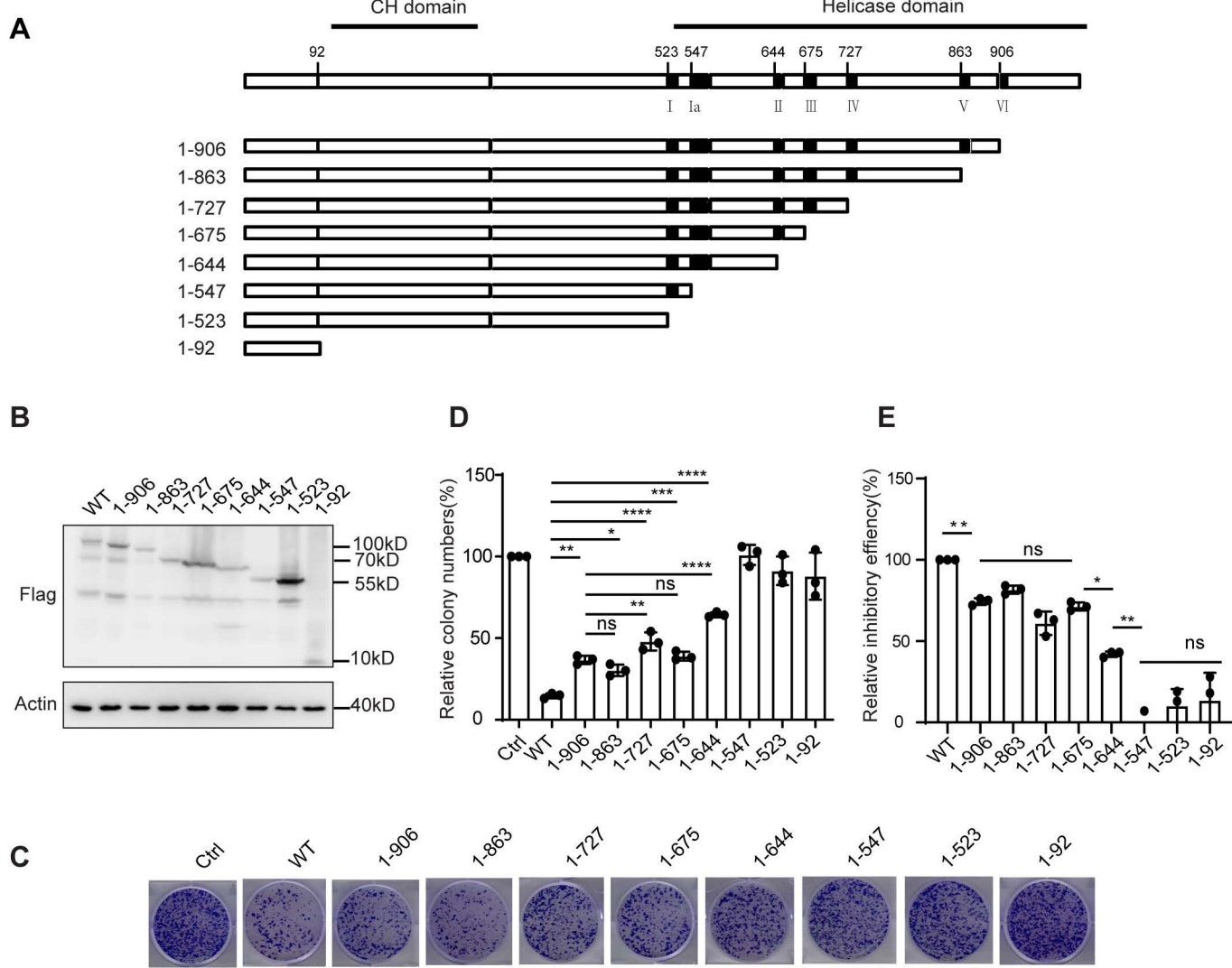

**Fig 1. Role of MOV10 C-terminal in its anti-LINE-1 activity.** (A) Schematic of the structure of MOV10 C-terminal deletion mutants. Numbers indicate AA positions. The locations of the putative CH domain and seven helicase motifs are indicated. (B) HeLa cells were co-transfected with CMV-L1-neo$^{RT}$ plasmids expressing full length MOV10 or MOV10 C-terminal mutants. Cells were subjected to western blots and LINE-1 retrotransposition assay (n = 3 biological replicates). Western blots were probed with antibodies for the detection of MOV10 (anti-Flag) and Actin expression. (C) Cell colonies formed by HeLa cells transfected with CMV-L1-neo$^{RT}$ plasmids and MOV10 or MOV10 C-terminal mutants. The data from three independent experiments were summarized in the bar graph. (D) Relative cell colonies number of Fig 1C was quantified. The data from three independent experiments were summarized in the bar graph. Error bars indicate SD, P-value was determined using ordinary one-way ANOVA test. (E) Relative inhibitory activity of MOV10 C-terminal truncations compared with MOV10 wild type. The data from three independent experiments were summarized in the bar graph. Error bars indicate SD, P-value was determined using ordinary one-way ANOVA test. ns means no significance; *P < 0.05; **P < 0.01; ***P < 0.001; ****P < 0.0001.

Using a similar strategy, we constructed a series of N-terminal truncations of MOV10, as graphically illustrated in Fig 2A (Fig 2A). We first examined the expression levels of these MOV10 N-terminal truncations (Fig 2B), followed by assessing their inhibitory effects on LINE-1 retrotransposition using the cell colony assay described above (Fig 2C). The results showed that among the N-terminal truncations, 532-1003 and 563-1003 possessed LINE-1 similar restriction activity as intact MOV10 did, whereas the others completely lost the function (Fig 2D and 2E). This observation demonstrates that

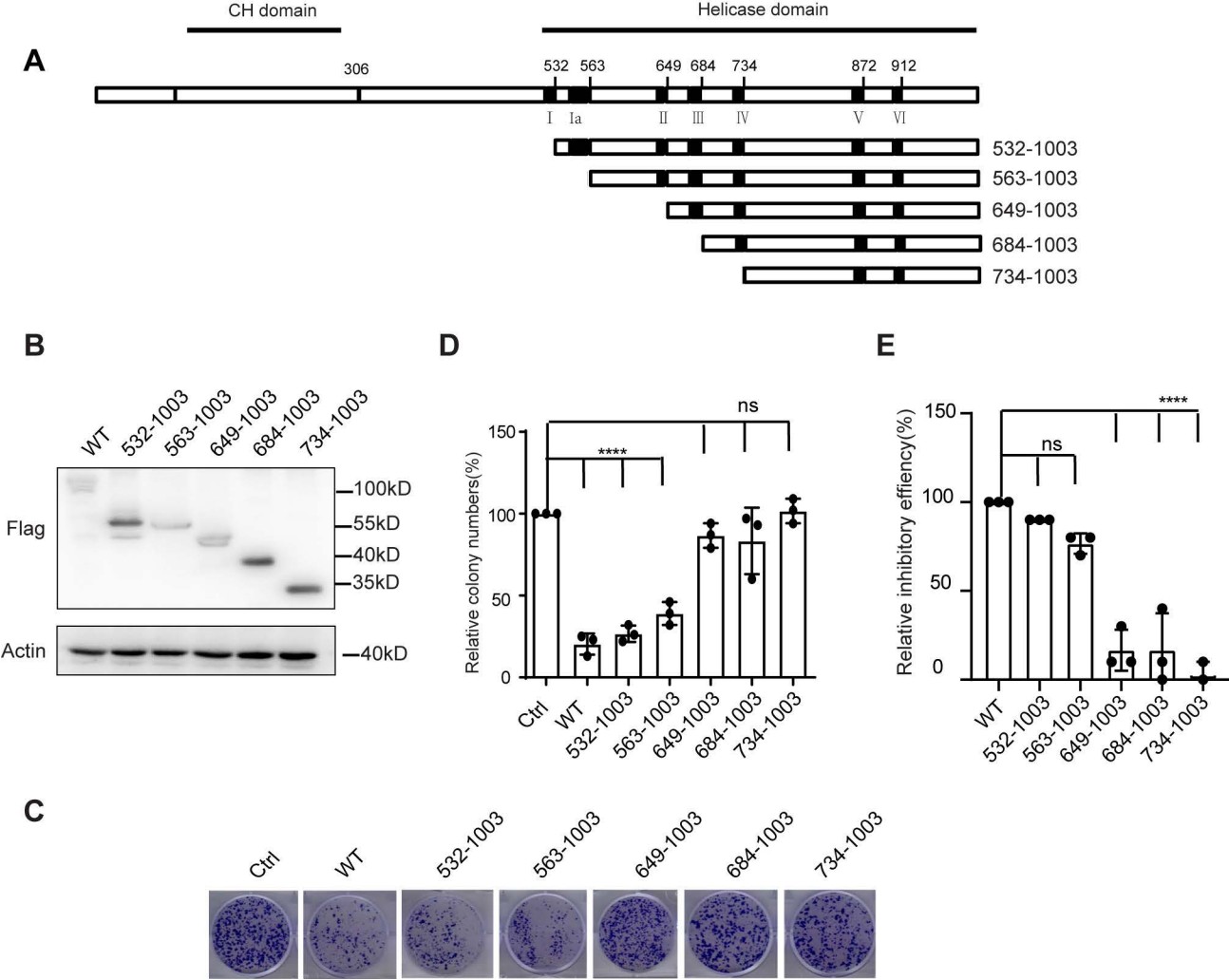

**Fig 2. Role of MOV10 N-terminal in its anti-LINE-1 activity.** (A) Schematic of the structure of MOV10 N-terminal deletion mutants. Numbers indicate AA positions. The locations of the putative CH domain and seven helicase motifs are indicated. (B) HeLa cells were co-transfected with CMV-L1-neo$^{RT}$ plasmids expressing full length MOV10 or MOV10 N-terminal mutants. Cells were subjected to RNA isolation, western blots, and LINE-1 retrotransposition assay. Western blots were probed with antibodies for the detection of MOV10 (anti-Flag) and Actin expression. (C) Cell colonies formed by HeLa cells transfected with CMV-L1-neo$^{RT}$ plasmids and MOV10 or MOV10 N-terminal mutants. (D) Relative cell colonies number of Fig 2C was quantified. The data from three independent experiments were summarized in the bar graph. Error bars indicate SD, P-value was determined using ordinary one-way ANOVA test. (E) Relative inhibitory activity of MOV10 N-terminal truncations compared with MOV10 wild type. The data from three independent experiments were summarized in the bar graph. Error bars indicate SD, P-value was determined using ordinary one-way ANOVA test. ns means no significance; ****P<0.0001.

the MOV10 N-terminus including motif Ia is almost dispensable for the anti-LINE-1 activity, despite a slight reduction activity observed upon the removal of motif Ia, e.g., 563-1003. Together with the result from C-terminal truncation analysis, we concluded that beside the C-terminus (amino acids 907-1003, referred as "**C-terminal domain**" thereafter), the domain containing motif II between motifs Ia and III (amino acids 563-675, referred as "**extended motif II**" thereafter) paly a vitally important role in blocking LINE-1 retrotransposition. Moreover, the latter is dominantly attributed to MOV10-mediated inhibitory effect on LINE-1 replication, since its removal cause a complete loss in the anti-LINE-1 function of MOV10, while the C-terminal domain of MOV10 is required for the maximal activity.

To minimize the influence of transfection efficiency and cell cytotoxicity from various MOV10 plasmids on the cell colony system, we employed a GFP co-transfection approach to monitor transfection efficiency (S1A Fig) and utilized the CCK8 assay to evaluate cell cytotoxicity. The results demonstrated that the transfection efficiency of GFP co-transfected with different MOV10 plasmids remained highly consistent (S1B Fig), and their cytotoxicity was negligible compared to the control group (S1C Fig).

## MOV10 mutants with anti-LINE-1 activity diminish the level of LINE-1 RNA

Our previous studies reported that MOV10 induced the degradation of LINE-1 RNA through a DCP2-mediated decapping mechanism, resulting in the restriction of LINE-1 replication [33]. Next, we investigated whether the anti-LINE-1 activity of the mutations tested above is results from decapping and degradation of LINE-1 RNA as previously reported. In agreement with early reports, expressing wild type MOV10 caused an approximately 80% reduction in LINE-1 RNA levels, compared with that of empty vector control (Fig 3A). Among the C-terminal truncations of MOV10, the presence of 1-906, 1-863, 1-727, 1-675 or 1-644 reduced LINE-1 RNA to about 40–65% of control group, while no effect was observed for remaining mutants, correlating with the anti-LINE-1 results shown in Fig 1. Similarly, among the N-terminal truncations tested above, only the overexpression of either 532-1003 or 563-1003 resulted in a reduction of LINE-1 RNA to approximately 35-50% of control group (Fig 3B), both of which were found to be active against LINE-1 (Fig 2). These data showed that the abilities of these mutants to reduce LINE-1 RNA were well correlated with their inhibitory effects on LINE-1 retrotransposition.

Next, we further confirmed whether the reduction of LINE-1 RNA by these mutants was a result of the decapping of LINE-1 RNA. Anti-m$^7$G cap antibody was applied to pull down the capped RNA in RNA immunoprecipitation experiments of RNAIP, and qPCR was used to detect the capped LINE-1 RNA. To ensure input RNA levels were similar in the cells with or without MOV10 expression, we knocked down endogenous *XRN1* using siRNA to prevent the degradation of decapped RNA as previously described [33] (S2 Fig). The results showed that upon the expression of wild type MOV10 or its mutants able to cause the reduction of LINE-1 RNA, such as C-terminal truncation 1-906 and N-terminal truncation 563-1003, capped LINE-1 RNA was reduced to half the level of control group (Fig 3C and 3D). However, no significant change in capped LINE-1 RNA levels was observed in the presence of either C-terminal truncation 1-547 or N-terminal truncation 649-1003 (Fig 3D), both of which failed to reduce the level of LINE-1 RNA and inhibit LINE-1 replication (Figs 1 and 2).

To avoid the conformational effect of large truncated mutants, two MOV10 mutants, namely Δ563-675 and the double mutant Δ563-675/Δ907-1003, were constructed(Fig 3E). HeLa cells were then transfected with either full-length MOV10, the single mutant Δ907-1003 (1-906), the single mutant Δ563-675, or the double mutant Δ563-675/Δ907-1003, with the L1-CMV-neo$^{RT}$ vector. The results revealed that both the Δ563-675 mutant and the double mutant Δ563-675/Δ907-1003 lost their anti-LINE-1 activity. In contrast, the Δ907-1003 mutant retained partial inhibitory activity against LINE-1, which is consistent with the result of MOV10 truncations (Fig 3F and 3G).

These data together suggest that the MOV10 mutants inhibit LINE-1 retrotransposition through the same mechanism as wild type form. In addition, this provides further evidence that MOV10 inhibits LINE-1 replication by diminishing RNA levels through LINE-1 RNA decapping, and that the C-terminal domain and the extended motif II of MOV10 play a vitally important role in the process.

## The ability of the mutated MOV10 to form cellular granules and co-localize with LINE-1 ORF1p

MOV10 was reported to associate with both LINE-1 RNP and DCP2, and subsequently form large cytoplasmic granules via LLPS, which is required for efficient LINE-1 mRNA decay [33]. To understand how the C-terminal domain and the extended motif II of MOV10 co-operate to maximize anti-LINE-1 activity, we thus used a cell imaging approach to investigate whether MOV10 mutants can induce the formation of cytoplasmic granules and co-localize with LINE-1 ORF1p.

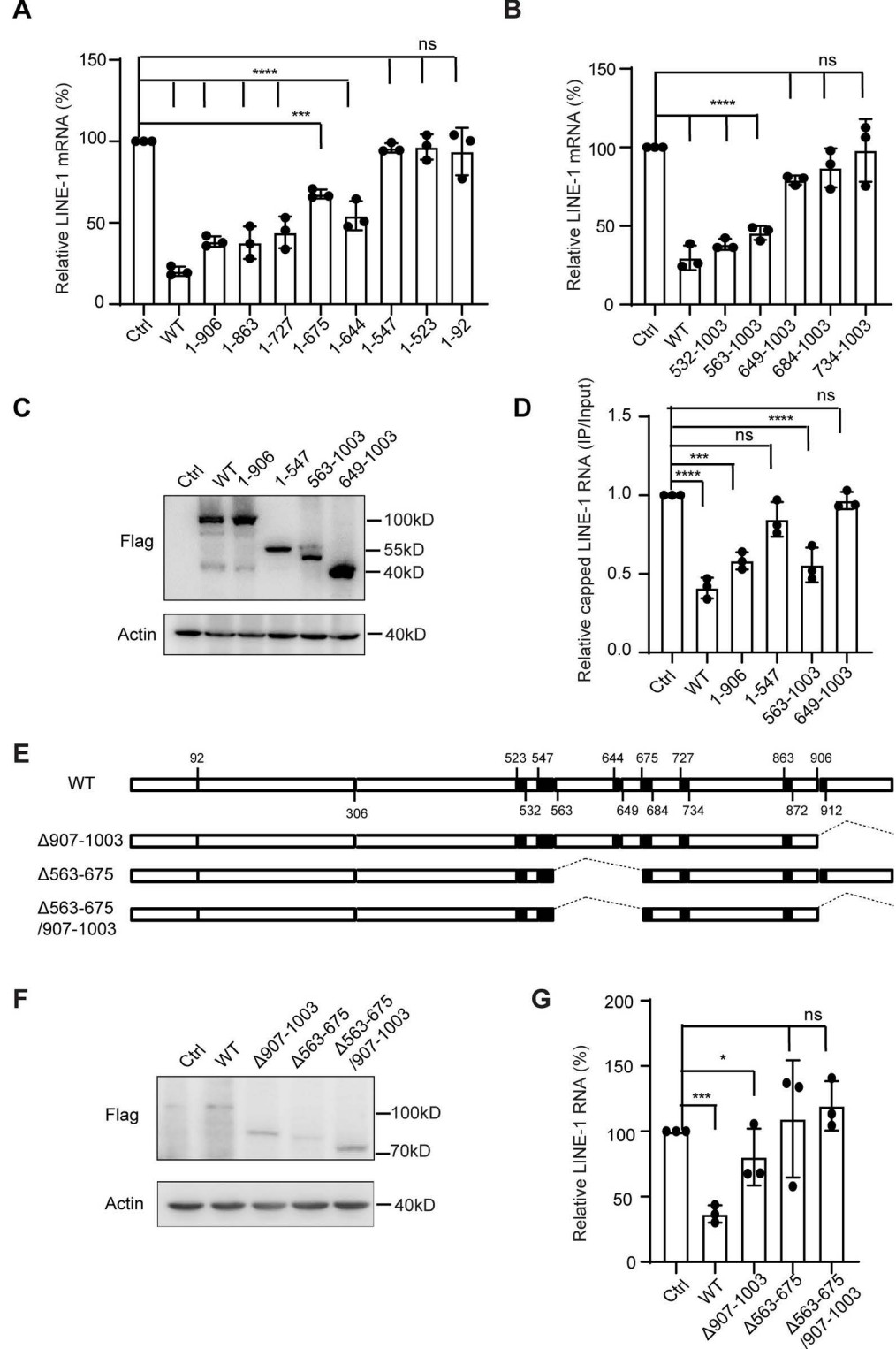

**Fig 3. Effect of MOV10 mutants on LINE-1 RNA level.** (A) RNA isolated from HeLa cells from Fig 1 was quantified by RT–qPCR and normalized to GAPDH expression. The data from three independent experiments were summarized in the bar graph. Error bars indicate SD, P-value was determined using ordinary one-way ANOVA test. (B) RNA isolated from HeLa cells from Fig 2 was quantified by qPCR and normalized to GAPDH expression. The

data from three independent experiments were summarized in the bar graph. Error bars indicate SD, P-value was determined using ordinary one-way ANOVA test. (C) Western blots were probed with antibodies for the detection of MOV10 (anti-Flag) and Actin expression. (D)The level of capped LINE-1 RNA in the presence of MOV10 and truncations. The HEK293T cells were treated with XRN1 specific siRNAs, and then transfected with CMV-L1-neo$^{RT}$ and MOV10 or MOV10 truncations. The cell lysate was collected for RNA immunoprecipitation using m$^7$G-cap specific antibody. The input and immunoprecipitated RNA were quantified for LINE-1 RNA level by qPCR, and the ratio of immunoprecipitated LINE-1 RNA to total LINE-1 RNA represent the capped LINE-1 RNA level (n = 3 biological replicates). Error bars indicate SD, P-value was determined using ordinary one-way ANOVA test. (E)Schematic of the structure of MOV10 Δ907-1003, Δ563-675 and Δ563-675/Δ907-1003. Numbers indicate AA positions. (F)Western blots were probed with antibodies for the detection of MOV10, Δ563-675, Δ907-1003, Δ563-675/Δ907-1003 (anti-Flag) and Actin expression. (G) The level of LINE-1 RNA in the presence of MOV10 Δ563-675, Δ907-1003 and Δ563-675/Δ907-1003.Error bars indicate SD, P-value was determined using ordinary one-way ANOVA test. ns means no significance; *P < 0.05; ***P < 0.001; ****P < 0.0001.

MOV10 truncations were co-transfected with CMV-L1-neo$^{RT}$ in HeLa cells, followed by immunofluorescence staining. As shown in Fig 4A, wild type MOV10 appeared as large foci co-localized with LINE-1 ORF1p (top panel). Whereas all the C-terminal truncations of MOV10 are diffusely distributed in the cytoplasm, except for 1-92, which is mainly located in the nucleus. Once the C-terminus amino acids 907-1003 is removed, MOV10-containing granules almost disappeared in the presence of LINE-1, while some dispersed MOV10 likely co-localized with ORF1p (Fig 4A). This observation strongly suggests that the C-terminal domain contributes significantly to the formation of cytoplasmic granules, an important factor required for the full anti-LINE-1 activity of MOV10.

In line with our conclusion, all the N-terminal truncations were found to form the cellular granules (Fig 4B), suggesting the N-terminal part of MOV10 up to amino acid 532 is dispensable for the formation of cytoplasmic granules. Of note, once the sequence containing the extended motif II was deleted, the mutated MOV10 did not co-localize with LINE-1 ORF1p (Fig 4B), accompanied with smaller particle size than that of cellular granules containing both MOV10 and ORF1p (S3 Fig). This suggests that motif II plays a key role in the association of MOV10 and LINE-1, which is required for the formation of large granules, as previously reported [33]. Additionally, Imaging analysis demonstrated that removal of amino acids 563-675 resulted in the formation of substantially smaller cytoplasmic granules that showed an inability to co-localize with ORF1p. Furthermore, the double mutant Δ563-675/Δ907-1003 exhibited a complete loss of granule formation, instead displaying a diffuse cytoplasmic distribution pattern (Fig 4C). These observations are fully consistent with the characteristic distribution patterns observed for MOV10 truncations.

### The C-terminal domain of MOV10 is essential for its association with G3BP1 and granules formation

Ras-GTPase-activating protein SH3-domain-binding protein 1 (G3BP1) is considered as a molecular switch that triggers RNA-dependent LLPS [41]. In line with it, our early evidence showed that deletion of G3BP1 markedly diminished the formation of LLPS containing MOV10/LINE-1 and impaired MOV10-mediated anti-LINE-1 activity [33]. The recruitment of G3BP1 represents one of the mechanisms to initiate and promote the process of phase separation. Therefore, we supposed that deletion of the C-terminus of MOV10 may affect its interaction with G3BP1, thereby inhibiting the formation of cellular granules (Fig 4A). As shown in the results of co-immunoprecipitation, the interaction between MOV10 C-terminal truncations (1-906 and1-863) and G3BP1 was significantly reduced, while N-terminal truncations efficiently interact with G3BP1 as the wild type MOV10 did (Fig 5A), indicating a role of the C-terminus in facilitating the MOV10/G3BP1 association. Of note, N-terminal truncation 684-1003 was able to recruit G3BP1 (Fig 5A), but it was shown a complete loss of anti-LINE-1 activity (Fig 2). These results provided evidence supporting the importance of recruiting G3BP1 by the C-terminus of MOV10 for the granules formation (Fig 4) and maximal anti-LINE-1 activity (Fig 1), whereas association of MOV10 with G3BP1 alone is insufficient for the inhibition of LINE-1 retrotransposition.

To better define the role of interacting with G3BP1 in the anti-LINE-1 activity of MOV10, we next used the inactive N-terminal truncation but able to interact with G3BP1, to compete with the wild type MOV10 for the association with G3BP1, followed by monitoring LINE-1 replication through quantification of the LINE-1 RNA level (Fig 5B). As expected,

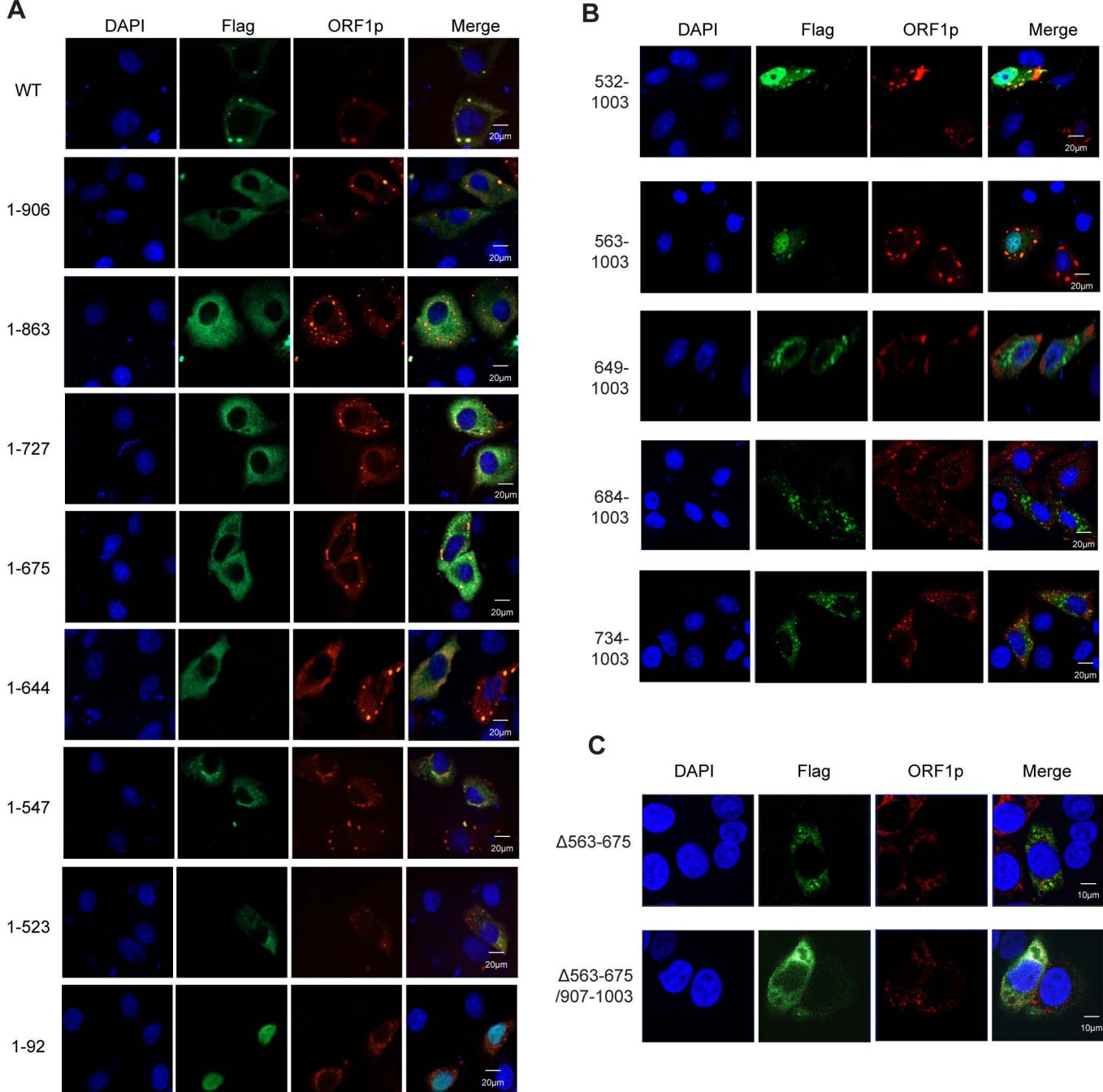

**Fig 4. The ability of the mutated MOV10 to form cellular granule and co-localize with LINE-1 ORF1p.** (A) Confocal microscopy showing cytoplasmic localization of C-terminal MOV10 mutants and ORF1p (n = 3 biological replicates). HeLa cells were transfected with CMV-L1-neo[RT] and C-terminal mutant plasmids. Immunofluorescence confocal microscopy was performed to determine the subcellular localization of DAPI (blue), ORF1p (Red) and MOV10(Green). Scale bars represent 20 μm. (B) Confocal microscopy showing cytoplasmic localization of N-terminal MOV10 mutants and ORF1p (n = 3 biological replicates). HeLa cells were transfected with CMV-L1-neo[RT] and N-terminal mutant plasmids. Immunofluorescence confocal microscopy was performed to determine the subcellular localization of DAPI (blue), ORF1p (Red) and MOV10(Green). Scale bars represent 20 μm. (C) Confocal microscopy showing cytoplasmic localization of MOV10 mutants Δ563-675 and the double mutant Δ563-675/Δ907-1003 with ORF1p (n = 3 biological replicates). HeLa cells were transfected with CMV-L1-neo[RT] Δ563-675 or Δ563-675/Δ907-1003. Immunofluorescence confocal microscopy was performed to determine the subcellular localization of DAPI (blue), ORF1p (Red) and MOV10(Green). Scale bars represent 10 μm.

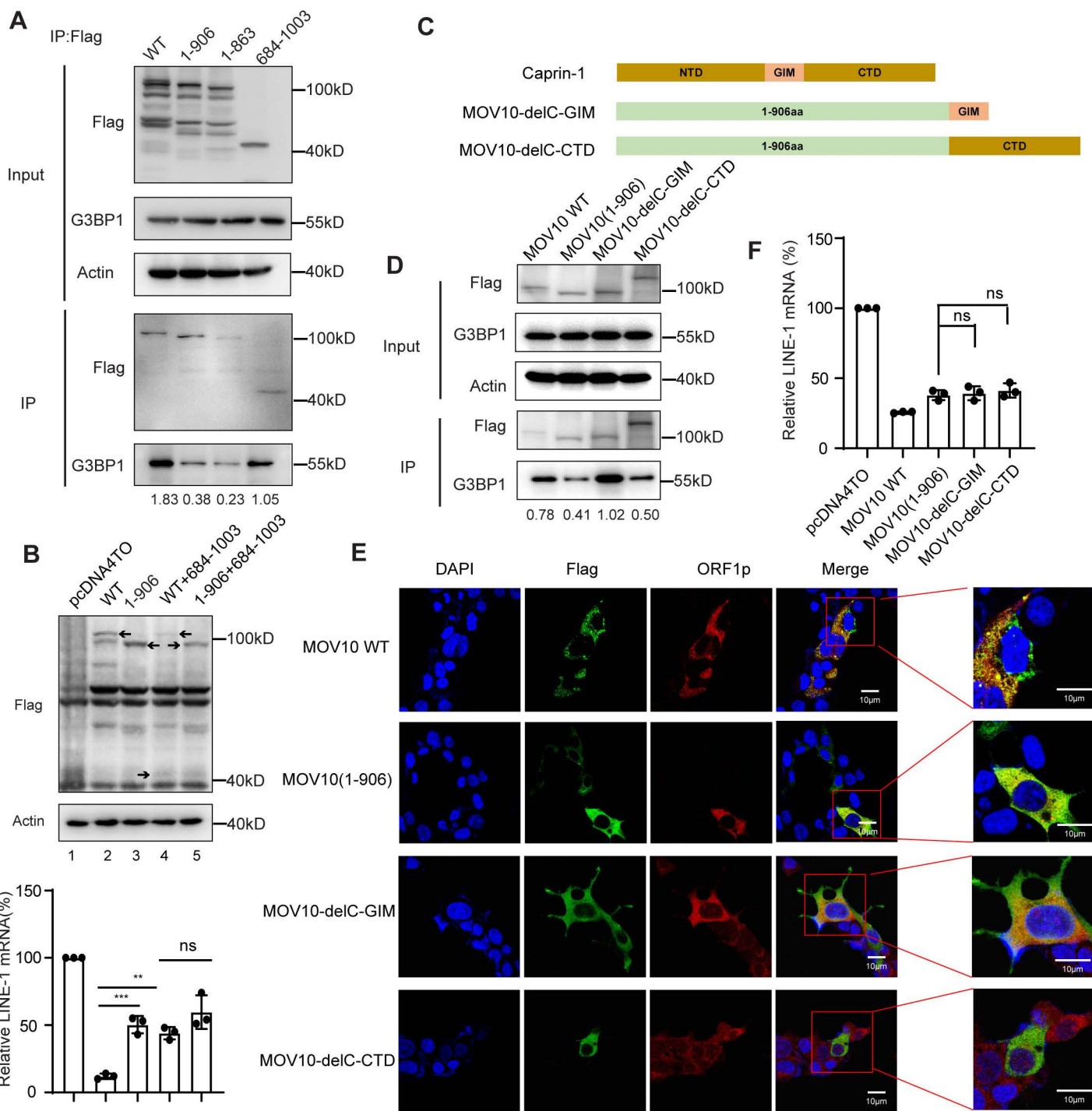

**Fig 5. The C-terminal domain of MOV10 is essential for association with G3BP1 and granules formation.** (A) Interactions among endogenous G3BP1 and MOV10 or MOV10 truncations. MOV10 was overexpressed in HEK293T cells. Input and anti-Flag IPs were subjected to immunoblot analysis using antibodies against G3BP1, Flag and Actin (n = 3 biological replicates). (B) HeLa cells were transfected with CMV-L1-neo^RT and full length MOV10 or 1-906 with or without N terminal truncation 684-1003. Western blots were probed with antibodies for the detection of MOV10 WT, 1-906,684-1003 (anti-Flag) and Actin expression. RNA isolated from HeLa cells was quantified by RT-qPCR and normalized to GAPDH expression. The data from three independent experiments were summarized in the bar graph. Error bars indicate SD, P-value was determined using ordinary one-way ANOVA test. (C) Schematic of two MOV10/Caprin-1 chimeric construct MOV10-delC-GIM and MOV10-delC-CTD. (D) Interactions among endogenous G3BP1 and

MOV10 or MOV10/Caprin-1 chimera. MOV10 or MOV10/Caprin-1 chimera was overexpressed in HEK293T cells. Input and anti-Flag IPs were subjected to immunoblot analysis using antibodies against G3BP1, Flag and Actin (n = 3 biological replicates). (E) Confocal images cytoplasmic localization of MOV10/Caprin-1 chimera and ORF1p using anti-Flag and anti-ORF1p antibodies. (F) Anti-LINE-1 activity of MOV10/ Caprin-1 chimera. HEK293T were co-transfected with CMV-L1-neo^RT and MOV10 or MOV10/Caprin-1 chimera. Cells were subjected to RNA isolation, and LINE-1 RNA was quantified by RT-qPCR and normalized to GAPDH expression. The data from three independent experiments were summarized in the bar graph. Error bars indicate SD, P-value was determined using ordinary one-way ANOVA test. ns means no significance, **P < 0.01, ***P < 0.001.

the results showed that the expression of 684-1003 reduced the anti-LINE-1 activity of full length MOV10 to a similar extent as 1-906 did, further supporting the important role of recruiting G3BP1 by the C-terminus of MOV10 in achieving maximal anti-LINE-1 activity (Fig 5B, lane3 and 4). In addition, the expression of N-terminal truncation containing the C-terminal domain 684–1003aa, failed to enhance the activity of the C-terminal truncation containing the motif II 1-906aa (Fig 5B, lane 5), indicating that the two domains of MOV10 shall co-operate in a cis manner.

To further explore the role of the C-terminal domain other than mediating the G3BP1/MOV10 association, we investigated the effect of the replacing MOV10 C-terminal domain with other G3BP1-binding domain upon the formation of cellular granules. A G3BP1-interacting-motif (GIM) was previously identified in Caprin-1, a stress granule component protein, and is involves in its binding to the NTF2L domain of G3BPs [42]. A chimeric construct was built, in which the C-terminal domain of MOV10 was substituted with the GIM motif (residues 369-378 of Caprin-1), named MOV10-delC-GIM (Fig 5C). As expected, the incorporation of GIM into the C-terminal truncation of MOV10 significantly restored the interaction between MOV10 and G3BP1, determined by Co-IP assay (Fig 5D). However, we found that MOV10-delC-GIM failed to form cellular granules (Fig 5E) or show better anti-LINE-1 activity than that of MOV10 C-terminal truncation without GIM (Fig 5F). These results suggest that besides binding to G3BP1, the C-terminal domain of MOV10 likely involve other functions required for promoting granules formation. Alternatively, it is possible that the way of binding G3BP1 by MOV10 C-terminus differs from that of GIM, thereby the latter fails to support the formation of cellular granules and the enhancement of anti-LINE-1 activity.

It is worthy noted that a disordered sequence (amino acids 966-1003) was predicted within the C-terminal domain published in Uniprot, which represents a key feature for promoting LLPS. To confirm the role of 966–1003aa in granules formation, the construct MOV10Δ966-1003 was used. The results demonstrated that deletion of the 966-1003aa region did not impair MOV10's ability to co-localize with ORF1p and form larger granules (S4A Fig), indicating that this region contributes minimally to the formation of large granules. qPCR data revealed that the Δ966-1003 mutant retained full anti-LINE-1 activity (S4B and S4C Fig), comparable to that of the full-length MOV10. These findings suggest that the C terminal amino acids 906–966 region plays a critical role in the formation of granules.

In addition, we also tested whether adding a disordered sequence to the C-terminal truncation is able to restore the ability to form cellular granules. The CTD of Caprin-1 was identified as a disordered sequence that underwent spontaneous LLPS [43]. A chimeric construct was built as described above, in which the C-terminal domain of MOV10 was substituted with the CTD of Caprin-1, named MOV10-delC-CTD (Fig 5C). However, these results showed that MOV10-delC-CTD failed to interact with G3BP1 (Fig 5D), form cellular granules (Fig 5E) or show better anti-LINE-1 activity than that of MOV10 C-terminal truncation (Fig 5F). This provides evidence supporting that the presence of a disordered sequence in MOV10 is insufficient for the formation of cellular granules.

## The extended Motif II is crucial for association of MOV10 and ORF1p

Previous studies have shown that the association with LINE-1 RNP is a key prerequisite for the regulation of LINE-1 RNA metabolism and retrotransposition by MOV10 [33]. Here, we found that once the extended motif II was deleted, MOV10 N-terminal truncations did not co-localize with LINE-1 ORF1p (Fig 4B), suggesting that the role of the extended motif II in MOV10-mediated anti-LINE-1 activity is to bind to LINE-1 RNP. To further address this, we examined the interaction

between MOV10 truncations and LINE-1 RNP using Co-IP assay. As shown in Fig 6A, among the C-terminal truncations, 1-906, 1-863, 1-727 or 1-675, and 1-644 were able to interact with ORF1p, while 1-547 and 1-523 showed no interaction. This suggests that amino acids 547-644 of MOV10 are required for its association with LINE-1 RNP. We also investigated the potential interaction between the nuclear-localized 1-92 and ORF1p, and no interaction was observed as expected (S5 Fig). Among the N-terminal truncations, only 532-1003 and 563-1003 were found to bind to ORF1p, suggesting that the region downstream of amino acid 563 is essential for the MOV10/LINE-1 interaction (Fig 6B). Taken together, these results demonstrated the important role of the extended motif II in mediating the interaction between MOV10 and LINE-1 RNP.

## Discussion

In this report, we explored the mechanism underlying MOV10-mediated inhibition of LINE-1 replication using a mutagenesis approach and found that the extended motif II (amino acids 563-675, between motifs Ia and III) and the C-terminal domain (amino acids 907-1003) of MOV10 co-operated to achieve maximal inhibition on LINE-1 retrotransposition. Mechanism studies revealed that the extended motif II is involved in the interaction between MOV10 and LINE-1, and the C-terminal domain is required for both the formation of cellular granules and the association of MOV10 with G3BP1. The association with LINE-1 through the extended motif II is dominantly attributed to MOV10-mediated anti-LINE-1 activity. On this basis, promoting granule formation by the C-terminal domain ensures the maximal inhibitory effect of LINE-1 replication by MOV10 (Fig 7). Consistent with our previous findings that G3BP1 plays a critical role in the LLPS-driven

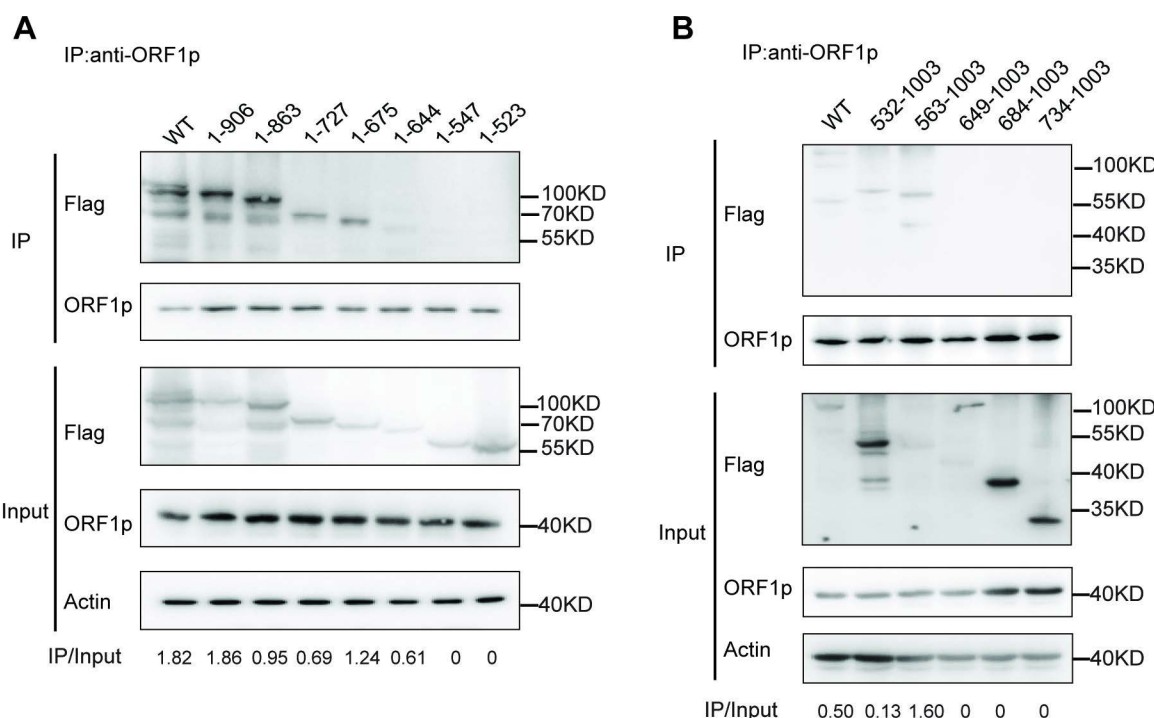

**Fig 6. The extended motif II is crucial for association of MOV10 and ORF1p.** (A) HEK293T cells were transfected with CMV-L1-neo[RT] and C-terminal mutant plasmids. The cell lysate was collected for immunoprecipitation using anti-ORF1p specific antibody. The IPs were subjected to immunoblot using antibodies against MOV10 and ORF1p (n = 3 biological replicates). (B) HEK293T cells were transfected with CMV-L1-neo[RT] and N-terminal mutant plasmids. The cell lysate was collected for immunoprecipitation using anti-ORF1p specific antibody. The IPs were subjected to immunoblot using antibodies against MOV10 (anti-Flag) and ORF1p (n = 3 biological replicates).

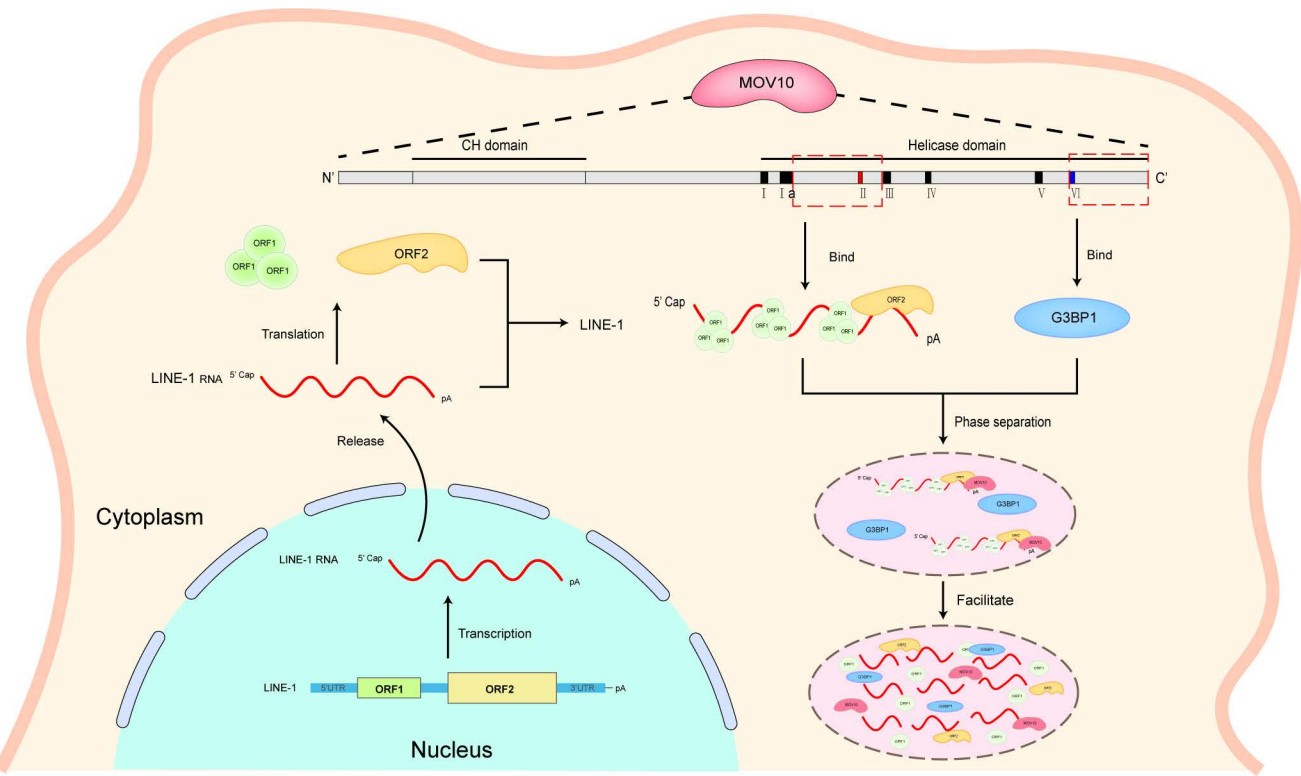

**Fig 7. Model of the restriction of LINE-1 retrotransposition by MOV10.** Associating with LINE-1 RNP and G3BP1 are essential for MOV10's anti-LINE-1 activity. Extend motif II (amino acids 563-675) of MOV10 is essential for its combination with LINE-1 RNP, and C-terminal domain (amino acids 907-1003) is crucial for associating with G3BP1 to form granules.

formation of MOV10/LINE-1 granules and the inhibition of LINE-1 retrotransposition [33], here we found that association of MOV10 with G3BP1 to form large granules is essential to retain maximal inhibitory activity on LINE-1.

This study found that the association with LINE-1 through the extended motif II is dominantly attributed to MOV10-mediated anti-LINE-1 activity. In agreement with the conclusion, we previously reported that the inactive MOV10EQ containing a E647Q in helicase motif II, loses its ability to associate with ORF1p [33]. These data together demonstrate the important role of helicase motif II in the association of MOV10 with LINE-1. It is worth noting that another inactive MOV10KR containing a K530R in helicase motif I was able to interact with LINE-1 ORF1p, but unable to bind to DCP2, accompanied by a loss of anti-LINE-1 activity [33]. Accordingly, besides binding to LINE-1 RNP, the extended motif II likely contribute to the ability of MOV10 to inhibit LINE-1 in other aspects such as the recruitment of DCP2.

G3BP proteins are essential for LLPS granules assembly [41], and cellular proteins such as Caprin-1 [44], USP10 [45] as well as OGFOD1 [46], bind with G3BP1 to regulate the dynamic assembly and disassembly of SGs. In line with it, we herein found that deletion of the C-terminus of MOV10 diminished its interaction with G3BP1 (Fig 5), accompanied by the impaired formation of cellular granules (Fig 4A). Of note, the replacement of MOV10 C-terminal domain with G3BP1-interacting-motif (GIM) of Caprin-1, significantly restored the binding of MOV10 to G3BP1, but failed to form cellular granules (Fig 5E) or enhanced the anti-LINE-1 activity of MOV10 C-terminal truncation (Fig 5F). These results suggested that association of MOV10 with G3BP1 is insufficient for the granule formation, and the C-terminal domain of MOV10 shall involve other functions required for promoting LLPS. Indeed, early studies showed that Caprin-1 GIM was not sufficient for G3BP1-driven granules [43].In addition, the replacement of MOV10 C-terminal domain with a disordered sequence within

Caprin-1 CTD also failed to form cellular granule or enhance the anti-LINE-1 activity of MOV10. Although the detailed mechanism how MOV10 induces LINE-1-containing LLPS awaits further investigation, this work provides further evidence to support that promoting LLPS by the C-terminal domain warrant maximal inhibitory effect of LINE-1 replication by MOV10.

According to the MOV10 structure published on UniProt, the region 966–1003 is predicted to be a disordered sequence, which is one of the critical factors for the formation of phase-separated structures. Our findings demonstrated that deletion of amino acids 966-1003 in MOV10 does not compromise its functional capacity. Specifically, the truncated MOV10 variant maintains the ability to form large cytoplasmic granules and complete activity comparable to the full-length MOV10(S4 Fig), suggesting an importance role of amino acids 906-966 in puncta formation. Among the many characterized phase-separating proteins, common molecular features, such as multivalency [47], intrinsically disordered regions [41], and structured RNA-binding [48] have been demonstrated to play critical roles in driving condensation. The interaction between RNA-binding proteins and RNA represents another pivotal element in the assembly of phase-separated granules. Prior research has demonstrated that RNA facilitates the formation of LLPS granules [49], yet an overabundance of RNA can result in their disintegration [48]. The C-terminal region (amino acids 907-1003) of MOV10 is presumed to be instrumental in RNA binding, orchestrating the formation of phase-separated granules via its engagement with RNA and a deeper exploration is required to elucidate the exact mechanisms at play.

Apparently, the subcellular distribution of MOV10 did not significantly affect its activity against LINE-1. For example, although 532-1003 and 563-1003 were predominantly located in the nucleus, both of them were co-localized with LINE-1 ORF1p in cytoplasmic granules and efficiently inhibited LINE-1 retrotransposition (Fig 2). This observation suggests a rapid recognition of LINE-1 by MOV10 occurred before its nuclear import.

The study allowed us propose a model that MOV10 helicase motif II associate with LINE-1 RNP and motif VI associate with G3BP1 to drive formation of LLPS in the meantime. When MOV10 recognize LINE-1 RNP, it may recruit many cell factors concurrently, including not just scaffolding protein such as G3BP1, TIA-1 to form LLPS, but also certain LINE-1 restriction factors such as TUT4 [20], RNASEH2 [21] and DCP2 [33] to induce LINE-1 RNA degradation. Previous data showed that TUT4 also enriched in cytoplasmic foci [20], and SAMHD1 causes sequestration of LINE-1 ORF1p into large cytoplasmic granules [19], providing a hypothesis that MOV10 may act as a platform to recruit other factors, inducing LINE-1 RNA degradation. Formation of LLPS driven by MOV10 may enclose series factors together to restrict LINE-1 replication via different mechanism.

## Materials and methods

### Plasmids and antibodies

The pcDNA4.0-based MOV10 DNA clone encodes an N-terminal Flag-tagged human MOV10 protein. MOV10 terminal mutants 1-906aa, 1-863aa, 1-727aa, 1-675aa, 1-644aa, 1-547aa, 1-523aa, 1-92aa, 532-1003aa, 563-1003aa, 649-1003aa, 684-1003aa and 734-1003aa were also cloned into pcDNA4.0 with a Flag tag in N-terminal. CMV-L1-neo$^{RT}$ carries a complete human LINE-1 DNA copy and a neomycin resistance gene inserted just before the 3' UTR of LINE-1 in the opposite direction to the LINE-1 coding sequence. The neomycin resistance gene is inactivated by the presence of a forward intron, which can be removed during RNA splicing, thus producing a functional neomycin resistance gene after reverse transcription and integration.

Flag-MOV10-delC-GIM, Flag-MOV10-delC-CTD, Flag-MOV10Δ966-1003, Flag-MOV10Δ563-675 and Flag-MOV10Δ563-675/907-1003 were cloned into pcDNA4.0. Lipofectamine 2000 (Invitrogen) was used for transient transfection of plasmids into HEK293T cells. Lipofectamine 3000 (Invitrogen) was used for transient transfection of plasmids into HeLa cells. Anti-Flag antibody (mouse) and anti-Flag antibody (rabbit) was purchased from Sigma-Aldrich and anti-Actin antibody (mouse) were purchased from Abcam. ORF1p antibody (rabbit) was generated as previously described [23]. Alexa Fluor 647-labeled donkey anti-goat antibody, Alexa Fluor 555-labeled donkey anti-rabbit antibody, Alexa

Fluor 488-labeled donkey anti-mouse antibody, Alexa Fluor 555-labeled donkey anti-mouse antibody, and Alexa Fluor 488-labeled donkey anti-rabbit antibody were purchased from Life Technologies.

## Cell culture

Human embryonic kidney HEK293T cells and HeLa cells were grown at 37°C in Dulbecco's modified Eagle's medium (DMEM; Gibco) supplemented with 10% fetal bovine serum (Gibco) in a humidified incubator at 5% $CO_2$.

## Co-immunoprecipitation (Co-IP) assay

HEK293T cells were co-transfected with CMV-L1-neo[RT] and Flag-MOV10 DNA or Flag-MOV10 mutant DNA. The cells were collected 48h post-transfection and then lysed in 350 µL of TNT buffer (20 mM Tris-HCl, pH 7.5, 200 mM NaCl, 1% Triton X-100) on ice for 1h with gentle rotation. The insoluble material was pelleted at 12,000 $g$ for 30 min and the supernatant was transferred into a new tube. Small aliquots from each sample were saved as "input", and remaining lysates were incubated with 5 µL anti-ORF1p antibody for 16h at 4 °C, followed by the addition of protein A+G-Sepharose (Amersham Biosciences) for 2h. The immunoprecipitated complex was then washed three times using TNT buffer and phosphate-buffered saline, followed by western blot analysis using anti-Flag or anti-ORF1p antibody.

## Quantification of LINE-1 RNA by RT-qPCR

HeLa cells transfected with plasmid were collected 48h later, and total RNA was extracted using the Rapure Total RNA kit (Magen Biotechnologies). cDNA was synthesized using Moloney murine leukemia virus (MMLV) reverse transcriptase (Takara), followed by treatment with DNase (Takara). cDNA was quantified using a qPCR kit (Sso Fast Eva Green Supermix, Takara) using the primers 5'-CTGAAGCGGGAAGGGACTG-3' and 5'- CCTTGAGCCTGGCGAACAG-3', which were designed to target to neo gene span the Neo cassette intron of the transfected LINE-1 construct such that only LINE-1 cDNA that has been reverse transcribed from the spliced RNA is amplified to avoid the contamination by CMV-L1-neo[RT] DNA.

## Western blotting

Cells were lysed with NP-40 buffer (Beyotime). Equal amounts of cell lysate were separated by SDS-PAGE (10%). Proteins were transferred onto a PVDF membrane, blocked with 5% skimmed milk, and probed with primary antibodies, including anti-Flag antibody (diluted 1:5,000), anti-ORF1p antibody (diluted 1:1,000), or anti-Actin antibody (diluted 1:5,000) at 4 °C overnight. After washing four times using PBS plus 0.1% Tween 20 (PBST), the membrane was incubated with a 1:5,000 dilution of HRP-conjugated goat-anti-mouse secondary antibody for 1h at room temperature. After washing four times using PBST, signals were detected using Western Lighting Chem Illustrine Science Reagent.

## RIP

HEK293T cells were transfected with 1,000 ng CMV-L1-neo[RT] with 500 ng Flag-MOV10 or MOV10 mutants. The cells were collected 48h post-transfection and then lysed in 350 µL of TNT buffer (20 mM Tris-HCl, pH 7.5, 200 mM NaCl, 1% Triton X-100) on ice for 1h with gentle rotation. The insoluble material was pelleted at 12,000 g for 30 min and the supernatant was transferred into a new tube. Small aliquots from each sample were saved as "input", and remaining lysates were incubated with 5 µL of anti- m[7]G-cap antibody or anti-IgG antibody for 16h at 4 °C, followed by the addition of protein A+G Sepharose (Amersham Biosciences) for 2h. The immunoprecipitated complex was then washed three times using TNT buffer and phosphate-buffered saline, followed by RNA extraction using the Rapure Total RNA kit (Magen Biotechnologies). The input and immunoprecipitated RNAs were quantified for LINE-1 RNA level by RT-qPCR. The immunoprecipitated LINE-1 RNA was normalized to input LINE-1 mRNA.

## Retrotransposition assay

HeLa cells were seeded in 6-well plates 1 day prior to transfection. The next day, cells were co-transfected with 1,000 ng CMV-L1-neo^RT DNA and Flag-MOV10 or MOV10 mutants. Forty-eight hours later, cells were detached from the plates using trypsin and split for western blot, RNA isolation, and the retrotransposition assay. Cells for the retrotransposition assay were seeded into 6-well plates at serial dilutions ($2\times10^5$ per well), and G418 (0.4mg/mL) was then added to select for resistant cell colonies. After 10–12 days of selection, when cell colonies were clearly visible, the cells were fixed with methanol for 10 min and stained with 0.5% crystal violet (in 25% methanol) for 10 min. The number of colonies represented the transposition efficiency of LINE-1.

## Immunofluorescence microscopy

Cells were incubated in a glass bottom cell culture dish (Nest) before transfection with CMC-L1-neo^RT and MOV10 or MOV10 mutants. Forty-eight hours after transfection, cells were fixed with 4% paraformaldehyde (in 1x phosphate buffered saline) for 15 min at room temperature followed by a 10 min permeabilization using 0.2% TritonX-100 at room temperature. Cells were then incubated for 1h with anti-Flag antibody (diluted 1:5,000) and anti-ORF1p antibody (diluted 1:200), followed by Alexa Fluor 647-labeled donkey anti-goat antibody, Alexa Fluor 555-labled donkey anti-rabbit antibody, and Alexa Fluor 488-labeled donkey anti-mouse antibody (1:1,000 dilution). Confocal images were acquired at room temperature using an Olympus IX81 Microsystem.

## Reagent or Resource

Reagent or resource used in experiments were listed in the table (Table 1).

## Supporting information

**S1 Fig. Transfection efficiency and potential toxic side effects of the exogenous MOV10 plasmids.** (A) HeLa cells were co-transfected with 500ng full length MOV10 or MOV10 mutants and 500ng GFP plasmid. Immunofluorescence confocal microscopy was performed to determine the subcellular localization of DAPI (blue), MOV10(Red) and GFP(Green). (B)Transfection efficiency of GFP transfected with MOV10 or MOV10 mutants. Error bars indicate SD, P-value was determined using ordinary one-way ANOVA test. (C). HeLa cells were transfected with 500ng full length MOV10 or MOV10 mutants, CCK-8 assays were performed to assess cell proliferation at 48h. The data from three independent experiments were summarized in the bar graph. Error bars indicate SD, P-value was determined using ordinary one-way ANOVA test. ns means no significance.
(TIF)

**S2 Fig. Endogenous XRN1 was knocked down by siRNA.** Western blots were probed with antibodies for the detection of XRN1and Actin expression.
(TIF)

**S3 Fig. Granules size of full length MOV10 and MOV10 mutants.** Granules size of full length MOV10 and MOV10 mutants. The area of granules formed by full-length MOV10,649-1003,684-1003 and 734-1003 (n=49,66,77,55). Error bars indicate SD, P-value was determined using ordinary one-way ANOVA test. ****P<0.0001.
(TIF)

**S4 Fig. Cytoplasmic localization of MOV10 Δ966–1003 and its anti-LINE-1 activity.** (A)Confocal images cytoplasmic localization of MOV10 Δ966-1003 and ORF1p using anti-Flag and anti-ORF1p antibodies. (B) Western blots were probed with antibodies for the detection of MOV10 WT, Δ966-1003 (anti-Flag) and Actin expression. (C)RNA isolated from HeLa

**Table 1.** Reagent or resource used in the experiments.

| REAGENT or RESOURCE | SOURCE | IDENTIFIER |
|---|---|---|
| **Antibodies** | | |
| Monoclonal anti-Flag M2 antibody | Sigma-Aldrich | F3165 |
| Rabbit monoclonal [EPR20018-251] to DDDDK tag | abcam | ab205606 |
| Goat polyclonal to DDDDK tag | abcam | ab1257 |
| Rb pAb to MOV10 | abcam | ab80613 |
| Anti-DCP2/TDT | abcam | ab28658 |
| G3BP1(H-10) Mouse monoclonal IgG | Santa cruz | sc-365338 |
| Anti-XRN1 antibody produced in rabbit | Sigma-Aldrich | SAB4200028 |
| Mouse monoclonal [mAbcam 8226] to beta Actin | abcam | Ab8226 |
| Anti-m3G, m7G-cap,clone H-20(mouse monoclonal) | Merck Millipore | cat#2912041 |
| Anti-ORF1p (Rb) | Gifted by Professor Guo Fei | |
| **Chemicals, Peptides, and Recombinant Proteins** | | |
| Lipofectamine 2000 | Thermo Fisher Scientific | 11668019 |
| Lipofectamine 3000 | Thermo Fisher Scientific | L3000015 |
| **Experimental Models: Cell Lines** | | |
| HEK293T | ATCC | CRL-11268 |
| HeLa | ATCC | CCL2 |
| **Oligonucleotides** | | |
| Primer:Flag-MOV10-delC-CTD Forward 1: GACTCTAGCGTTTAAACTTAAGCTTATGGATTACAAGGACGACGATGACAAGCCCAGTAAGTTCAGCTGCCGG | This paper | N/A |
| Primer: Flag-MOV10-delC-CTD Reverse 1: GGCAGGATCAAGTGTCTGATTTTCAAAATCCCCTCCATTGAACCTCTTGGGGTTCT | This paper | N/A |
| Primer: Flag-MOV10-delC-CTD Forward 2: AGAACCCCAAGAGGTTCAATGGAGGGGATTTTGAAAATCAGACAC | This paper | N/A |
| Primer: Flag-MOV10-delC-CTD Reverse 2: CCACCACACTGGACTAGTGGATCCTTACCCTCCACGACCTCGTGGGGCTCCC | This paper | N/A |
| Primer: Flag-MOV10-delC-GIM Forward: CCAAGCTTATGGATTACAAGGACGACGATGACAAGCCCAGTAAGTTCAGCTGCCGGCA | This paper | N/A |
| Primer: Flag-MOV10-delC-GIM Reverse: GCGGGATCCTTACAGCATTGAATCCTGTATGAAATTATAGGGATTGAACCTCTTGGGGT | This paper | N/A |
| Primer: Flag-MOV10 C-terminal truncations Forward: CCAAGCTTATGGATTACAAGGACGACGATGACAAGCCCAGTAAGTTCAGCTGCCGGCA | This paper | N/A |
| Primer: Flag-MOV10(1-906) Reverse: GCGGGATCCTTAATTGAACCTCTTGGGGTTCTTAAGGA | This paper | N/A |
| Primer: Flag-MOV10(1-863) Reverse: GCGGGATCCTTAACCCACCTTCAAGTCCTTGATGT | This paper | N/A |
| Primer: Flag-MOV10(1-727) Reverse: GCGGGATCCTTACTTGGTTATGAACTGGGGGTC | This paper | N/A |
| Primer: Flag-MOV10(1-675) Reverse: GCGGGATCCTTACAGCTGCCCTCCTGGATCACCT | This paper | N/A |
| Primer: Flag-MOV10(1-644) Reverse: GCGGGATCCTTAGATGAAGATGTGTGTGAAGTGA | This paper | N/A |
| Primer: Flag-MOV10(1-547) Reverse: GCGGGATCCTTAAGCGCAGGCCAAGATGTGGGCT | This paper | N/A |

*(Continued)*

| REAGENT or RESOURCE | SOURCE | IDENTIFIER |
|---|---|---|
| Primer: Flag-MOV10(1-523) Reverse: GCGGGATCCTTAAAAGATGATGTAGGGGGCTGG | This paper | N/A |
| Primer: Flag-MOV10(1-92) Reverse: GCGGGATCCTTATCTCCTCTTTTCTGGGAACCGCA | This paper | N/A |
| Primer: Flag-MOV10(532-1003) Forward: CCAAGCTTATGGATTACAAGGACGACGATGACAAGACGTTAGTGGAGG-CAATTAAG | This paper | N/A |
| Primer: Flag-MOV10(563-1003) Forward: CCAAGCTTATGGATTACAAGGACGACGATGACAAGTGTCAAAGGCTCCGG-GTCCACCTTCC | This paper | N/A |
| Primer: Flag-MOV10(649-1003) Forward: CCAAGCTTATGGATTACAAGGACGACGATGACAAGGCATGGAGCCTGA-GAGTCTGGTAGCTAT | This paper | N/A |
| Primer: Flag-MOV10(684-1003) Forward: CCAAGCTTATGGATTACAAGGACGACGATGACAAGGGGCCTGTGCTGC-GTTCCCCACTGACCC | This paper | N/A |
| Primer: Flag-MOV10(734-1003) Forward: CCAAGCTTATGGATTACAAGGACGACGATGACAAGCATCCCACCATCCTG-GACATTCCTA | This paper | N/A |
| Primer: Flag-MOV10 N-terminal truncations Reverse: GCGGGATCCTTAGAGCTCATTCCTCCACTCTGGCTCCAC | This paper | N/A |
| Primer: Flag-Δ563-675 Forward 1: AGCGTTTAAACTTAAGCTTATGGATTACAAGGATGACGACGATAAGCCCAGTA | This paper | N/A |
| Primer: Flag-Δ563-675 Reverse 1: TCTCCTGCCAGCACTAGGTCAGCCCCTGAGTTG | This paper | N/A |
| Primer: Flag-Δ563-675 Forward 2: CAACTCAGGGGCTGACCTAGTGCTGGCAGGAGA | This paper | N/A |
| Primer: Flag-Δ563-675 Reverse 2: CACTGGACTAGTGGATCCTTAGAGCTCATTCCTCCAC | This paper | N/A |
| Primer: Flag-Δ563-675/Δ907-1003 Forward 1: AGCGTTTAAACTTAAGCTTATGGATTACAAGGATGACGACGATAAGCCCAGTA: | This paper | N/A |
| Primer: Flag-Δ563-675/Δ907-1003 Reverse: CACTGGACTAGTGGATCCTTAATTGAACCTCTTGGGG | This paper | N/A |
| Primer: Flag-MOV10Δ966-1003 Forward 1: AGCGTTTAAACTTAAGCTTATGGATTACAAGGATGACGACGATAAGCCCAGTA | This paper | N/A |
| Primer: Flag-MOV10Δ966-1003 Reverse 1: CACTGGACTAGTGGATCCTTACAGACCTTGCAGTA | This paper | N/A |
| **Recombinant DNA** | | |
| Plasmid:Flag-MOV10 | XiaoyuLi.,2013 | N/A |
| Flag-MOV10(1-906) | This paper | N/A |
| Flag-MOV10(1-863) | This paper | N/A |
| Flag-MOV10(1-727) | This paper | N/A |
| Flag-MOV10(1-675) | This paper | N/A |
| Flag-MOV10(1-644) | This paper | N/A |
| Flag-MOV10(1-547) | This paper | N/A |
| Flag-MOV10(1-523) | This paper | N/A |
| Flag-MOV10(1-92) | This paper | N/A |
| Flag-MOV10(532-1003) | This paper | N/A |
| Flag-MOV10(563-1003) | This paper | N/A |
| Flag-MOV10(649-1003) | This paper | N/A |

*(Continued)*

**Table 1.** (Continued)

| REAGENT or RESOURCE | SOURCE | IDENTIFIER |
|---|---|---|
| Flag-MOV10(684-1003) | This paper | N/A |
| Flag-MOV10(734-1003) | This paper | N/A |
| Flag-MOV10-delC-CTD | This paper | N/A |
| Flag-MOV10-delC-GIM | This paper | N/A |
| Flag-MOV10Δ563-675 | This paper | N/A |
| Flag-MOV10Δ563-675/907-1003 | This paper | N/A |
| Flag-MOV10Δ966-1003 | This paper | N/A |
| CMV-L1-neo^RT | XiaoyuLi.,2013 | N/A |
| Image J | Image J software | https://imagej.nih.gov/ij/ |
| GraphPad Prism 8 | GraphPad software | N/A |

cells was quantified by qPCR and normalized to GAPDH expression. The data from three independent experiments were summarized in the bar graph. Error bars indicate SD, P-value was determined using ordinary one-way ANOVA test. ns means no significance.
(TIF)

**S5 Fig. Interactions among MOV10 or MOV10 1–92 and ORF1p in HEK293T cells.** Input and anti-Flag IPs were subjected to immunoblot analysis using antibodies against Flag, ORF1p and Actin. (n = 3 biological replicates).
(TIF)

**S1 Data. Source data with graph values.** Data used to generate graphs in all figures.
(XLSX)

**S2 Data. Source data for Western blots.** Source data for western blots in all figures.
(ZIP)

## Acknowledgments

We thank the National Infrastructure of Microbial Resources (NIMR-2014-3) and CAMS Collection Center of Pathogenic Micro-organisms (CAMS-CCPM-A) for providing valuable reagents.

## Author contributions

**Conceptualization:** Qian Liu, Yaqi Liu, Shan Cen.

**Data curation:** Qian Liu, Yaqi Liu, Xiaoyu Li.

**Formal analysis:** Yang Mao, Dongrong Yi, Quanjie Li, Ling Ma.

**Funding acquisition:** Qian Liu, Xiaoyu Li, Shan Cen.

**Investigation:** Jiwei Ding, Jing Wang.

**Methodology:** Qian Liu, Xiaoyu Li, Shan Cen.

**Project administration:** Xiaozhong Peng, Xiaoyu Li.

**Resources:** Dongrong Yi, Jiwei Ding, Saisai Guo.

**Software:** Yang Mao, Quanjie Li.

**Supervision:** Yongxin Zhang, Jianyuan Zhao.

**Validation:** Qian Liu, Saisai Guo.

**Visualization:** Qian Liu, Yaqi Liu.

**Writing – original draft:** Qian Liu.

**Writing – review & editing:** Xiaozhong Peng, Xiaoyu Li, Shan Cen.

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
