## [Decision Letter · Decision Letter 0]

6 Feb 2025

PGENETICS-D-24-01503

Maximal inhibitory effect of MOV10 on LINE-1 retrotransposition requires both the MOV10/LINE-1 association and phase separation

PLOS Genetics

Dear Dr. Cen,

Thank you for submitting your manuscript to PLOS Genetics. After careful consideration, we feel that it has merit but does not fully meet PLOS Genetics's publication criteria as it currently stands. Therefore, we invite you to submit a revised version of the manuscript that addresses the points raised during the review process.

Please submit your revised manuscript within 30 days Mar 08 2025 11:59PM. If you will need more time than this to complete your revisions, please reply to this message or contact the journal office at plosgenetics@plos.org. Please include the following items when submitting your revised manuscript:

We look forward to receiving your revised manuscript.

Kind regards,

Edward Chuong

Academic Editor

PLOS Genetics

Monica Colaiácovo

Section Editor

PLOS Genetics

Aimée Dudley

Editor-in-Chief

PLOS Genetics

Anne Goriely

Editor-in-Chief

PLOS Genetics

**Additional Editor Comments :**

Your paper has now been reviewed by 4 reviewers, who all appreciated the significance of the study, but collectively they had multiple important concerns. Reviewer 1 notes that the major claim in the title (“Phase separation”) is not adequately supported by the text. Reviewer 2 appreciates the deletion strategy, but feels the novelty is limited (especially with reference to missing citations Goodier et al PLOS Genetics 2012) and that the conclusions were not adequately supported by the data. Some of these issues may be addressed by more precisely defining terms such as “maximal” inhibitory effect and being more precise in discussing foci and phase separation. However, additional evidence that transfection efficiency was controlled for seems important. On a similar vein, reviewer 3 notes that key results in the G3BP1 gel in Fig 5A are relatively faint and would benefit from evidence of replicates. Reviewer 4 suggests several additional mutants which may also strengthen the controls in the study. Multiple reviewers noted that the manuscript was missing key references as well as labeling of statistics on figures.

Based on these reviews, we feel this study would be of interest to the readers of PLOS Genetics, but requires additional revisions to strengthen and clarify key claims. We would welcome a revised manuscript addressing the majority of the reviewer concerns.

**Journal Requirements:**

At this stage, the following Authors/Authors require contributions: Qian Liu, Yaqi Liu, Yang Mao, Dongrong Yi, Quanjie Li, Jiwei Ding, Saisai Guo, Yongxin Zhang, Jing Wang, Jianyuan Zhao, Ling Ma, Xiaozhong Peng, Xiaoyu Li, and Shan Cen. Please ensure that the full contributions of each author are acknowledged in the "Add/Edit/Remove Authors" section of our submission form.

The list of CRediT author contributions may be found here: https://journals.plos.org/plosgenetics/s/authorship#loc-author-contributions

https://journals.plos.org/plosgenetics/s/submission-guidelines#loc-parts-of-a-submission

- ® on page: 24.

5) Please upload a copy of Figure 6c which you refer to in your text on page 15. Or, if the figure is no longer to be included as part of the submission please remove all reference to it within the text.

Potential Copyright Issues:

i) Figure 7. Please confirm whether you drew the images / clip-art within the figure panels by hand. If you did not draw the images, please provide (a) a link to the source of the images or icons and their license / terms of use; or (b) written permission from the copyright holder to publish the images or icons under our CC BY 4.0 license. Alternatively, you may replace the images with open source alternatives. See these open source resources you may use to replace images / clip-art:

7) We note that your Data Availability Statement is currently as follows: "All data required for evaluating the conclusions of this study are available within the paper, or from the authors upon request.". Please confirm at this time whether or not your submission contains all raw data required to replicate the results of your study. Authors must share the “minimal data set” for their submission. PLOS defines the minimal data set to consist of the data required to replicate all study findings reported in the article, as well as related metadata and methods (https://journals.plos.org/plosone/s/data-availability#loc-minimal-data-set-definition).

8) Please amend your detailed Financial Disclosure statement. This is published with the article. It must therefore be completed in full sentences and contain the exact wording you wish to be published.

2) If any authors received a salary from any of your funders, please state which authors and which funders..

**Reviewers' comments:**

Reviewer's Responses to Questions

Reviewer #1: This is an interesting and important study that seeks to understand how the RNA helicase MOV10 suppresses LINE-1 retrotransposition. LINEs are the only active autonomous mobile DNA in the human genome, thus, their regulation is important to study. Creation of the MOV10 constructs, their use in the retrotranposition assay, the LINE-1 RNA degradation experiments and expression in cells made for a detailed and compelling study. The work identified two important regions in MOV10, “extended motif II” and the distal C-terminus (aa 907-1104) that cooperate to achieve maximal LINE 1 inhibition. Their role in puncta formation and colocalization with ORF1 was very compelling. Some of the conclusions were not correct, though, and must be addressed. There were also some mistakes and omissions that need to be fixed, as described below.

Major points.

1. There is a problem with the title: this paper does not formally show phase-separation, rather it demonstrates puncta/granule formation. It seems the title should be changed to reflect that.

2. Figure 1. the legend does not match the panels: C and E are switched

3. In general, all figure legends need to include the statistical tests used and the number of replicates tested.

4. The statement describing Fig. 4B says that the puncta are of smaller size. The sizes need to be quantified.

5. Please give the citation supporting the sentence on pg. 18 “It is worthy noted that a disordered sequence (amino acids 966-1003) was predicted within the C-terminal domain, which represents a key feature for promoting LLPS.”

6. The authors need to cite the following papers as showing evidence that MOV10 regulates LINE retrotransposition:

Goodier JL, Cheung LE, Kazazian Jr HH. MOV10 RNA helicase is a potent inhibitor of retrotransposition in cells. PLoS Genet. 2012;8(10):e1002941.

Skariah, G, Seimetz, J, Norsworthy, M, Lannom, MC, Kenny, PJ, Elrakhawy, M, Forsthoefel, C, Drnevich, J, Kalsotra, A, Ceman, S. (2017). Mov10 suppresses retroelements and regulates neuronal development and function in developing brain. BMC Biology. 15(1):54 PMID:28662698

7. In Figure 5C, there is a “CIM” domain in the schematic while the text refers to the Caprin G3BP1 interacting domain as “GIM.” This is confusing. Please rectify this in either the text of figure legend.

8. A statement in the Discussion is not correct: “These results suggest

that association of MOV10 with G3BP1 is required but insufficient for the granule formation, and the C-terminal domain of MOV10 shall involve other functions required for promoting LLPS.” There was no evidence in this manuscript that association with G3BP1 is required for MOV10 to be found in puncta. If there is evidence elsewhere, it should be cited here. In fact, the opposite seems to be true: when the authors engineered MOV10 with the GIM/CIM motif and forced association with G3BP1 (Fig. 5D), it did not lead to puncta formation (Figure 5E). What this work does show is that the C-terminal region from 906 to 1004 is required for MOV10 to form puncta, which is interesting and should be discussed.

9. Also in the Discussion that is not correct: because the authors see the N-terminal 92 amino acids of MOV10 in the nucleus, they infer that it has an NLS. That is incorrect: proteins less than 45 kDa in size can diffuse into the nucleus. This statement needs to be amended or removed.

10. There are a number of grammatical errors including a paucity of articles (the, a, etc..) and incorrect verb tenses. This should be corrected.

Reviewer #2: In the manuscript ”Maximal inhibitory effect of MOV10 on LINE-1 retrotransposition requires both the MOV10/LINE-1 association and phase separation”, the authors investigate how MOV10 inhibits L1 retrotransposition by employing the use of MOV10 deletion mutants. Their data suggests that the MOV10 extended motif II (563-675aa) and the CTD (906-1003aa) cooperate to achieve “maximal” MOV10-mediated inhibition of L1 activity. They propose that motif II is involved in mediating interactions between MOV10 and L1, and that the CTD is required for the association of MOV10 with G3BP1 and the formation of liqid-liqid phase separation (LLPS), which promotes the “maximal” inhibition of L1 activity by MOV10.

The use of MOV10 deletion mutants is logical strategy to approach the difficult problem of unraveling how MOV10 inhibits L1 retrotransposition. However, the findings of this study do not provide significant insight into this mechanism beyond what has already been reported. In addition, some of the main conclusions are not well supported by the data and some of the experiments lack important controls. The manuscript also lacks experimental details and is poorly referenced. Please see below for detailed comments:

1. It is unclear what the authors mean by “maximal” inhibitory effect. The term “maximal” is not defined clearly and therefore confusing. The data show that several of the MOV10 CTD mutants inhibit L1 (figure 1) to a significant effect in their assays. For example, MOV10 mutant 1-863 inhibition is ~80% compared to WT MOV10, which is indeed a very strong inhibitory effect. In addition, other CTD mutants still inhibited L1 quite strongly. Notably, the CTD mutants dd not form cytoplasmic foci (LLPS?), which suggests that foci formation is not strictly required for inhibition. Together, these data do not seem consistent with the title and the author’s model that the MOV10 CTD/G3BP1 interaction and LLPS is required for MOV10-mediated inhibition.

2. The authors did not indicate how they monitored transfection efficiency or how they monitored for potential toxic side effects of the transfected MOV10 plasmids in their retrotransposition assays. Variations in transfection efficiencies and/or toxicity of the MOV10 plasmids could result in differences in retrotransposition efficiency (for example see Fig. 1C and 1D) and/or variations in protein expression levels for the different MOV10 plasmids (e.g., Fig. 1B). The authors must monitor transfection efficiency and also monitor for potential toxic side effects of the exogenous MOV10 plasmids in their retrotransposition assays.

3. The manuscript is not adequately referenced; many factual statements do not have references or contain the wrong reference. Here are several of examples from the introduction although there are more instances throughout the manuscript:

a. Line 48: ORF1 chaperone activity should have at least the following references: (DOI: 10.1128/MCB.21.2.467-475.2001 and DOI: 10.1073/pnas.0809964106)

b. Line 53: “TPRT” requires reference to Luan et al. (DOI: 10.1016/0092-8674(93)90078-5)

c. The CMV-L1-neoRT plasmid should be referenced and attributed to the Heidmann lab (DOI: 10.1038/74184).

d. The author should at a minimum cite Moran et al, (DOI: 10.1016/s0092-8674(00)81998-4) for the retrotransposition assay.

e. Line 66: Goodier et al. should be cited for the fact that MOV10 inhibits L1 retrotransposition (DOI: 10.1371/journal.pgen.1002941)

4. The authors propose that MOV10 interaction with G3BP1 and LLPS formation is involved in inhibition of L1 retrotransposition. Notably, Goodier et al. (DOI: 10.1371/journal.pgen.1002941) first showed that ORF1p and MOV10 localize to cytoplasmic granules suggesting that cytoplasmic foci or granules could be involved in MOV10 inhibition. Subsequent studies examining other host factors that inhibit L1 have reported similar phenomenon, although the connection remains unclear between inhibition and cytoplasmic foci or granules. Since the authors propose that G3BP1 is involved in these foci, and somehow connected with L1 inhibition, it would be helpful to know the proportion of ORF1p/MOV10 foci that actually contain G3BP1. Also, one could ask the question whether MOV10 inhibits L1 in cells lacking G3BP1 or whether ORF1p/MOV10 foci form in cells that lack G3BP1. Do the authors think the foci they are seeing are stress granules?

5. Throughout the manuscript figures, there is no indication of how many times experiments were repeated and what statistics were used to determine significance (p-values). The authors need to indicate how many times experiments were repeated and how results were calculated as well as the statistical methods used to analyze results if they are going to show p-values.

6. In figure 1 it is unclear what the bars and asterisks indicate in the graphs in 1D and 1E.

7. The source of the ORF1p antibody should be listed in the methods.

8. Please indicate MW size markers (kDa) on all western blot images.

Reviewer #3: This is an interesting and thorough study of how MOV10 restricts LINE-1 (L1) retrotransposition in human cells. Testing a series of MOV10 C-terminus and N-terminus mutants in retrotransposition assays, the authors nicely show which parts of the MOV10 helicase inhibit LINE-1 mobility and interact with G3BP1, which has shown be shown to be involved in phase separation. The presence of LINE-1 ORF1p in granules, which has been studied extensively in relation to MOV10, is shown here to rely on the G3BP1 interacting C-terminus of MOV10. The data also suggest that the extended motif II of the MOV10 helicase interacts with the LINE-1 RNP (although it isn't defined for certain whether this is more closely with ORF1p or ORF2p). When combined, interactions between the extended motif II and the C-terminus of the MOV10 helicase with the LINE-1 RNP and G3BP1, respectively, are necessary for MOV10 to exert its maximal effect as an LINE-1 inhibitor. These conclusions are reasonable based on the data presented. Generally I found the presentation was clear and I could follow the text easily. Although questions remain, this work makes a valuable advance on our understanding of how MOV10 inhibits LINE-1.

Moderate issue:

Figure 5A - how reproducible is the gel showing interactions with G3BP1. Although I agree with the interpretation, the bands are relatively faint. It would be useful in supplemental to show two more replicates of this experiment. I realise that this will take additional work but it seems it important to show what parts of the MOV10 helicase are interacting with G3BP1.

Minor issues:

line 53 - please cite PMID: 1722352 alongside the Feng et al EN paper.

line 58 - I would cite a review summarising diseases caused by LINE-1 insertions (e.g. PMID: 27158268 or a more recent one).

line 67 - mentioning DNA methylation and other factors limiting LINE-1 transcription in human cells, I think it would be prudent to cite PMID: 31230816 and PMID: 38309261 here.

line 73 - it would be collegiate to cite the previous work from Goodier et al on MOV10 (PMID: 23093941)

Results - from the outset it could be stated that the retrotransposition assays were done in HeLa cells (although I do note that is said in the figures)

Nomenclature - interactions with ORF1 are noted where the authors are referring to the protein ORF1p.

Figure 5B - the histogram doesn't really line up with the labels of the blot above (assume x-axis has those labels).

line 375 - the Discussion uses the terminology "full" and "maximal" to describe LINE-1 inhibition. This is somewhat open to interpretation because the data show "maximal" inhibition is achieved, not "full" as there is still significant mobility despite the presence of untruncated MOV10.

Discussion - there are some mentions of previous studies that are uncited in the text; please provide these references.

Methods - please provide a reference for the CMV-L1-neoRT repoter.

Geoff Faulkner (University of Queensland)

Reviewer #4: This study was designed based on previous research that human MOV10 recruits DCP2 to decap LINE-1 RNA through liquid-liquid phase separation (LLPS), thereby inhibiting LINE-1 retrotransposition. In this study, the authors further explored the functional domains of MOV10 for LINE-1 inhibition and underlying mechanisms of MOV10 function. They found that MOV10 binds to LINE-1 through its extended motif II (563-675aa), which is the primary mechanism by which it inhibits LINE-1 activity. Furthermore, the C-terminal domain of MOV10 (906-1003aa) enhances LLPS, thereby maximizing MOV10's inhibitory effect on LINE-1 replication. This is an interesting and valuable study.

Major Points:

1.The study emphasizes the crucial role of MOV10's extended motif II (563-675aa) and C-terminal in LINE-1 inhibition. However, the authors did not directly truncate or construct these two regions separately or combined for further investigation, leading to a lack of more direct evidences. In addition, to avoid the conformational effect of large truncated mutants, the authors can construct plasmids of MOV10 mutant (Δ563-675aa), MOV10 mutant (Δ563-675aa and Δ906-1003 ) to verify their effects on LINE-1.

2. In Figure 5, when exploring the function of MOV10's C-terminal domain, the authors mention the disordered region (966-1003aa) in line 310 of the main text. It might be better to truncate this disordered region for further study.

3. In Figure 6A, the immunoprecipitation (IP) results show that pulled down MOV10 wild-type (lane 1) is significantly less than the truncated proteins (lanes 2-6), suggesting that the interaction between MOV10 WT with ORF1 is weaker than that of truncated mutants, which is odd. Moreover, they didn’t examine truncated 1-92, so it is just to infer its interaction with ORF1. Additionally, in Figure 6B, the input results show considerable differences in protein expression levels among the truncated mutants (lanes 3-4 are notably weaker). Better to quantify the bands by ImageJ. Similar issues appear in other figures such as Figure 2B, which might affect the results.

Minor Points:

1. In Figure 1D (line 143) and Figure 1E (line 144), the descriptions are reversed.

2. In Figure 3D, the first lane for XRN1 is present, but the first lanes for FLAG and Actin are missing. It would be better to maintain consistency.

3. In Figure 5B, the non-specific bands are too strong. Arrows should be used to indicate the positions of the MOV10 truncations in the figure.

4. The intensity of MOV10 (WT) and its truncations binding to G3BP1 in Figures 5A and 5D can be quantified using ImageJ.

**Have all data underlying the figures and results presented in the manuscript been provided?**

Reviewer #1: Yes

Reviewer #2: Yes

Reviewer #3: Yes

Reviewer #4: Yes

PLOS authors have the option to publish the peer review history of their article (what does this mean? ). If published, this will include your full peer review and any attached files.

**Do you want your identity to be public for this peer review?** For information about this choice, including consent withdrawal, please see our Privacy Policy .

Reviewer #1: No

Reviewer #2: No

Reviewer #3: **Yes: ** Geoffrey Faulkner

Reviewer #4: **Yes: ** Wenyan Zhang

**Figure resubmission:**
---

## [Decision Letter · Decision Letter 1]

10 Apr 2025

PGENETICS-D-24-01503R1

Maximal inhibitory effect of MOV10 on LINE-1 retrotransposition requires both the MOV10/LINE-1 association and granule formation

PLOS Genetics

Dear Dr. Cen,

Thank you for submitting your manuscript to PLOS Genetics. After careful consideration, we feel that it has merit but does not fully meet PLOS Genetics's publication criteria as it currently stands. Therefore, we invite you to submit a revised version of the manuscript that addresses the points raised during the review process.

Please submit your revised manuscript within 30 days May 10 2025 11:59PM. If you will need more time than this to complete your revisions, please reply to this message or contact the journal office at plosgenetics@plos.org. Please include the following items when submitting your revised manuscript:

We look forward to receiving your revised manuscript.

Kind regards,

Edward Chuong

Academic Editor

PLOS Genetics

Monica Colaiácovo

Section Editor

PLOS Genetics

Aimée Dudley

Editor-in-Chief

PLOS Genetics

Anne Goriely

Editor-in-Chief

PLOS Genetics

**Additional Editor Comments :**

The revised manuscript has been re-reviewed by the original reviewers. While reviewers 3 and 4 are satisfied with the revisions, reviewers 1 and 2 still have important concerns--particularly reviewer 1 who raises an issue with the Co-IP data presented in Figure 6. Please prepare and submit a revision that addresses their comments.

**Journal Requirements:**

At this stage, the following Authors/Authors require contributions: Qian Liu, Yaqi Liu, Yang Mao, Dongrong Yi, Quanjie Li, Jiwei Ding, Saisai Guo, Yongxin Zhang, Jing Wang, Jianyuan Zhao, Ling Ma, Xiaozhong Peng, Xiaoyu Li, and Shan Cen. Please ensure that the full contributions of each author are acknowledged in the "Add/Edit/Remove Authors" section of our submission form. 

The list of CRediT author contributions may be found here: https://journals.plos.org/plosgenetics/s/authorship#loc-author-contributions 

**Reviewers' comments:**

Reviewer's Responses to Questions

Reviewer #1: I am very confused by Figure 6: ORF1 appears to co-immunoprecipitate equally with all of the immunoprecipitated MOV10 proteins, regardless of whether they were even immunoprecipitated with the anti-Flag antibody. Specifically, the last three MOV10 truncations are not present in the Flag immunoprecipitation, although there is equal ORF1 co-immunoprecipitated. Unless there is a labeling problem, it appears that ORF1 non-specifically associates with the immunoprecipitating antibody. That control should be shown. Specifically, what happens when you immunoprecipitatate LINE-expressing HeLa cells without any Flag-tagged protein?

Further, Figure S3 does not support this statement “Our findings demonstrated that deletion of amino acids 966-1003 in MOV10 does not compromise its functional capacity. Specifically, the truncated MOV10 variant maintains the ability to form large cytoplasmic granules and complete activitycomparable to the full-length MOV10(S3 Fig).”

Figure S3 shows the granule sizes

Reviewer #2: The authors indicate that anti-Flag antibodies were used for experiments presented in Figure 6, but this does not seem correct. In the methods they indicate that they used either anti-Flag or anti-ORF1p antibodies. Please confirm which antibodies were used for the IP experiments shown in Fig. 6.

Reviewer #3: The authors have addressed my comments, thank you.

Reviewer #4: The authors have addressed my concerns.

**Have all data underlying the figures and results presented in the manuscript been provided?**

Reviewer #1: Yes

Reviewer #2: Yes

Reviewer #3: Yes

Reviewer #4: Yes

PLOS authors have the option to publish the peer review history of their article (what does this mean? ). If published, this will include your full peer review and any attached files.

**Do you want your identity to be public for this peer review?** For information about this choice, including consent withdrawal, please see our Privacy Policy .

Reviewer #1: No

Reviewer #2: No

Reviewer #3: **Yes: ** Geoffrey J Faulkner

Reviewer #4: **Yes: ** Wenyan Zhang

**Figure resubmission:**
---

## [Decision Letter · Decision Letter 2]

5 May 2025

Dear Dr Cen,

We are pleased to inform you that your manuscript entitled "Maximal inhibitory effect of MOV10 on LINE-1 retrotransposition requires both the MOV10/LINE-1 association and granule formation" has been editorially accepted for publication in PLOS Genetics. Congratulations!

Yours sincerely,

Edward Chuong

Academic Editor

PLOS Genetics

Monica Colaiácovo

Section Editor

PLOS Genetics

Aimée Dudley

Editor-in-Chief

PLOS Genetics

Anne Goriely

Editor-in-Chief

PLOS Genetics

Comments from the reviewers (if applicable):

Reviewer's Responses to Questions

**Comments to the Authors:**

Reviewer #1: I greatly appreciate the authors' corrections, which makes this manuscript important to the community

Reviewer #2: The authors have addressed my question regarding the antibodies used for IP experiments shown in Fig. 6.

**Have all data underlying the figures and results presented in the manuscript been provided?**

Reviewer #1: Yes

Reviewer #2: Yes

PLOS authors have the option to publish the peer review history of their article (what does this mean? ). If published, this will include your full peer review and any attached files.

**Do you want your identity to be public for this peer review?** For information about this choice, including consent withdrawal, please see our Privacy Policy .

Reviewer #1: No

Reviewer #2: No

**Data Deposition**

http://datadryad.org/submit?journalID=pgenetics&manu=PGENETICS-D-24-01503R2

**Press Queries**

---

## [Editor Report · Acceptance letter]

PGENETICS-D-24-01503R2

Maximal inhibitory effect of MOV10 on LINE-1 retrotransposition requires both the MOV10/LINE-1 association and granule formation

Dear Dr Cen,

We are pleased to inform you that your manuscript entitled "Maximal inhibitory effect of MOV10 on LINE-1 retrotransposition requires both the MOV10/LINE-1 association and granule formation" has been formally accepted for publication in PLOS Genetics! Your manuscript is now with our production department and you will be notified of the publication date in due course.

With kind regards,

Zsofia Freund

PLOS Genetics

On behalf of:
